



# Contrasts in dissolved, particulate and sedimentary organic carbon from the Kolyma River to the East Siberian Shelf

Dirk Jong[1], Lisa Bröder[1,2], Tommaso Tesi[3], Kirsi Keskitalo[1], Nikita Zimov[4], Anna Davydova[4], Philip Pika[1], Negar Haghipour[2], Timothy Eglinton[2], Jorien Vonk[1]

[1] Department of Earth Sciences, Vrije Universiteit, Amsterdam, the Netherlands
[2] Geological Institute, Swiss Federal Institute of Technology, Zürich, Switzerland
[3] Institute of Polar Sciences, National Research Council, Bologna, Italy
[4] Pacific Geographical Institute, Far East Branch, Russian Academy of Sciences, Northeast Science Station, Cherskiy, Russia

*Correspondence to*: Dirk Jong (d.j.jong@vu.nl), Jorien Vonk (j.e.vonk@vu.nl)

**Abstract.** Arctic rivers will be increasingly affected by the hydrological and biogeochemical consequences of thawing permafrost. During transport, permafrost-derived organic carbon (OC) can either accumulate in floodplain and shelf sediments or be degraded into greenhouse gases prior to final burial. Thus, the net impact of permafrost OC on climate will ultimately depend on the interplay of complex processes that occur along the source-to-sink system. Here, we focused on the Kolyma River, the largest watershed completely underlain by continuous permafrost, and marine sediments of the East Siberian Sea as

a transect to investigate the fate of permafrost OC along the land-ocean continuum. Three pools of riverine OC were investigated for the Kolyma main stem and five of its tributaries: dissolved OC (DOC), suspended particulate OC (POC), and riverbed sediment OC (SOC) and compared to earlier findings in marine sediments. Carbon isotopes ($\delta^{13}$C, $\Delta^{14}$C), lignin phenol, and lipid biomarkers show a contrasting composition and degradation state of these different carbon pools. Dual isotope source apportionment calculations imply that old permafrost-OC is mostly associated with sediments (SOC;

contribution of $68 \pm 10\%$), and less dominant in POC ($38 \pm 8\%$), while autochthonous primary production contributes around $44 \pm 10\%$ to POC in the main stem and up to $79 \pm 11\%$ in tributaries. Biomarker degradation indices suggest that Kolyma DOC is relatively degraded, regardless of its generally young age shown by previous studies. In contrast, SOC shows the lowest $\Delta^{14}$C signal (oldest OC), yet relatively fresh compositional signatures. Furthermore, decreasing mineral surface area-normalised OC- and biomarker loadings suggest that SOC is reactive along the land-ocean continuum supporting the idea that

floodplain and shelf sediments are efficient reactors. A better understanding of DOC and POC dynamics in Arctic rivers is still necessary, however, this study highlights that sedimentary dynamics play a crucial role when targeting permafrost-derived OC in aquatic systems. Chemical and physical processes (e.g. degradation, sorption) along fluvial-marine transects will determine to what degree thawed permafrost OC may be destined for long-term burial, therewith attenuating further global warming.

## 1 Introduction

Permafrost regions store approximately half of the global soil organic carbon (OC) (Hugelius et al., 2014; Zimov et al., 2006a). Amplified warming of the Arctic, currently three times as fast as the global average (IPCC, 2021), warms permafrost on a global scale (Biskaborn et al., 2019). Permafrost thaw and associated shifts in hydrology (Walvoord and Kurylyk, 2016), impact regional carbon cycling through the release of organic matter from this previously frozen pool to the fluvial network. In addition, the release of nutrients and sediment leads to a multitude of effects on the biogeochemical properties of inland and

coastal waters (Terhaar et al., 2021; Vonk et al., 2015). Furthermore, decomposition of OC from thawing permafrost soils releases greenhouse gases ($CO_2$, $CH_4$) into the atmosphere causing further climate warming (Schuur et al., 2015).

Arctic rivers, as rivers in general, serve as integrators of their catchments tracking changes in terrestrial signatures of the transported organic matter at the river mouth, and can therefore be used as indicators for watershed-wide processes such as permafrost thaw or soil remobilization (van Dongen et al., 2008; Wild et al., 2019; Feng et al., 2013). Based on river mouth

sampling campaigns, the six largest Arctic rivers are estimated to transport 40 Tg of fluvial OC, of which 34 Tg DOC and 6



Tg POC, into the Arctic Ocean (Holmes et al., 2012; McClelland et al., 2016). These estimates serve as important baseline data for terrestrial carbon export to the Arctic Ocean. However, fluvial OC cycling already occurs in headwater streams, and extends beyond the river mouth to the shelf seas. Inland waterways are known not just to conservatively channel fluvial OC towards the ocean, but also on one hand actively degrade OC into greenhouse gases and on the other hand sequester OC on

short- and long timescales (days to millennia) (Cole et al., 2007; Drake et al., 2018). Similarly, breakdown of terrestrial OC in the marine environment (e.g., Alling et al., 2010; Bröder et al., 2018), subsequent ocean acidification (Semiletov et al., 2016) and increase in marine primary production (Terhaar et al., 2021) have been the focus of recent studies. To better assess processing and fate of terrestrial organic matter in aquatic systems, we should regard these environments to be linked in a land-ocean continuum or as a carbon cycle 'without boundaries' (Battin et al., 2009).

For a complete assessment of fluvial OC, one needs to look at three different compartments: dissolved organic carbon (DOC; operationally defined as <0.7 µm), suspended particulate organic carbon (POC; >0.7 µm) and sedimentary organic carbon (SOC). In the six largest Arctic rivers, DOC concentrations are generally higher than those of POC (Holmes et al., 2012; McClelland et al., 2016), however, DOC consists predominantly of recent terrestrial material, while POC is predominantly sourced from deeper soils and permafrost (Wild et al., 2019). The fraction of DOC that is derived from Yedoma permafrost,

Pleistocene-aged permafrost deposits rich in OC, along the Kolyma River is rapidly degraded upon thaw (Mann et al., 2015; Rogers et al., 2021; Vonk et al., 2013). In contrast, POC derived from thermal erosion of river banks and coastlines, thermokarst, and other abrupt permafrost thaw features may be less prone to rapid degradation, and transported over longer distances (Keskitalo et al., 2022; Salvadó et al., 2016). Concentrations, fluxes and isotopic signatures of POC in Arctic rivers have been studied in the past decade (McClelland et al., 2016; Wild et al., 2019), including more recent studies on the molecular

structure and. degradation (e.g, Kolyma river; Bröder et al., 2020; Keskitalo et al., 2022). However, the cycling and degradation of POC during lateral aquatic transport, and especially its interplay with DOC and SOC remains elusive.

To better understand the interaction and exchange of POC with river- and marine sediments, as well as DOC, all these OC pools need to be considered. Yet to date, studies on riverine SOC transport and degradation are limited and contradictory. In the Danube River, SOC concentrations and mineral-specific surface area-normalised biomarker loadings decrease

downstream, suggesting significant SOC degradation during fluvial transport (Freymond et al., 2018). On the contrary, Scheingross et al. (2019) found in an experimental setting that particle abrasion and turbulent mixing of POC in the water column has only a limited effect on degradation, and suggests that degradation takes place mostly during floodplain storage of sediment. Repasch et al. (2021) (Rio Bermejo, Argentina) and (Hilton et al., 2015) (Mackenzie River, Canada) show that eroded POC is efficiently transported by rivers, and redeposited in floodplains or basins offshore, and suggest that sediment

transport time and mineral protection of OC regulate the magnitude and rate of POC degradation. Additionally, processes such as leaching of POC and SOC, and, vice versa, adsorption of DOC to soil or mineral particles influences both the composition and degradability of OC: mineral binding ballasts and slows down degradation of OC (Hemingway et al., 2019; Keil et al., 1994; Keskitalo et al., 2022; Kleber et al., 2021; Vonk et al., 2010b), while leaching of OC to the dissolved phase increases its potential for degradation (Abbott et al., 2014; Mann et al., 2015; Rogers et al., 2021; Vonk et al., 2013). No previous studies,

according to our knowledge, have addressed transport and degradation of SOC in the Kolyma River using riverbed samples upstream from the Kolyma river mouth.

Here we use an integrated approach, for a combined investigation of the dissolved, particulate and sediment fractions of OC along a 250-km long river transect in the lower reaches of the Kolyma River, including five of its tributaries. We apply a variety of bulk analyses (OC%, $\delta^{13}C$, $\Delta^{14}C$, mineral-specific surface area), and use molecular geochemical tracers (long-chain

*n*-alkanoic acids, and lignin and cutin-derived products) to untangle, for each fraction, the sources of OC and its degradation state. Furthermore, we connect our fluvial data with published records for a 1000 km-long transect across the East Siberian Sea (ESS) (Tesi et al., 2014; Vonk et al. 2010, 2012; Bröder et al., 2019; Salvadó et al., 2016), to track the changes of each



phase (dissolved, particulate and sediment OC) and the effect of fractionation and degradation on the state of terrestrial OC during aquatic transport over large distances along the land-ocean continuum.

**2 Methods**

**2.1 Study area and sample locations**

The Kolyma River in Northeast Siberia is the world's largest watershed (653,000 km$^2$) entirely underlain by continuous permafrost (Holmes et al., 2012). Its discharge follows a distinct seasonal pattern typical for Arctic rivers, with a strong peak during the spring freshet, and lower baseflow in winter. Annual water discharge is $109 \pm 7$ km$^3$ (Holmes et al., 2012), and the

average annual DOC and POC flux from the Kolyma River to the East Siberian Sea is 818 Gg ($10^9$ g) and 123 Gg per year, respectively (Holmes et al., 2012; McClelland et al., 2016). In its lower reaches, the river flows roughly northward through lowlands that consist of icy loess-like Yedoma deposits, or Ice complex permafrost deposits (ICD), of Pleistocene age. This Yedoma permafrost has a high OC content (2 – 5%; Zimov et al., 2006b). Most of the Kolyma watershed is covered by boreal forests (taiga) dominated by the Cajanderi larch (*Larix cajanderi Mayr*), and the Kolyma Delta further north is in the tundra

biome.

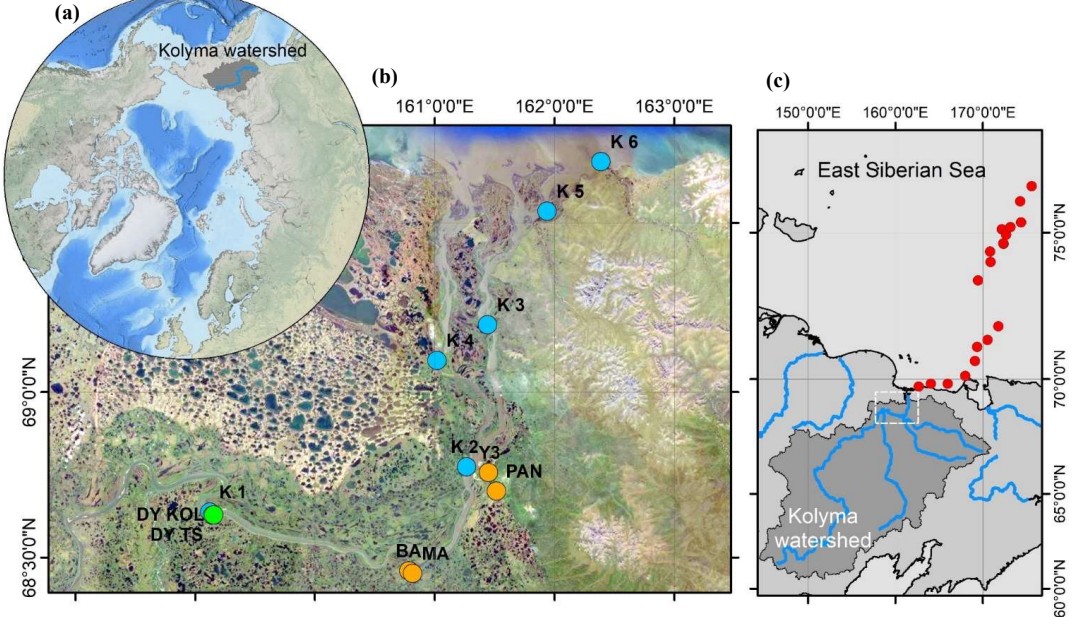

**Figure 1. (a) Location of the Kolyma watershed (Made with ArcMap™, Copyright © Esri. All rights reserved.). (b) Kolyma River delta with sample locations. In blue the Kolyma mainstem samples, orange the tributaries, green Duvanny Yar. For reference location K 2 is located at the town of Cherskiy. Background image adapted from (Mann et al., 2012). (c) Sample location in the East**
**Siberian Sea following the Kolyma paleoriver-transect, extended to the shelf break. The white box is the location of the Kolyma Delta. First 8 locations offshore from South to North: YS34B to YS41 (Vonk et al., 2010a; Tesi et al., 2014); 10 locations farther offshore from South to North: SWE-60, SWE-61, YS91, YS90, SWE-63, SWE-64, YS88, SWE-65, YS86, SWE-66 and SWE-67 (Salvadó et al., 2016; Bröder et al., 2019).**

Sampling of the Kolyma River took place from the Northeast Science Station in Cherskiy during summer 2018, from 23 July

to 3 August. We covered a 250 km-long transect of the Kolyma River starting at 68.63890 °N 159.12080 °E, where the river passes a ca. 10 km long Yedoma-deposit river bank exposure (Duvanny Yar, DY) to the delta outflow into the East Siberian Sea, including sampling the lesser-studied western delta branch of the Kolyma River (K1-K6, Fig. 1b, Table 1). In addition to the samples from the Kolyma River main stem, samples were taken from several tributaries with varying catchment sizes. Two of the larger tributaries of the Kolyma were sampled, the Maly Anyuy (MA), and the Bolshoy Anyuy (BA), with a catchment



size of 49,800 and 57,300 km², respectively, and a smaller tributary, the Panteleikha (PAN; 1,630 km²), where an algal bloom
       was observed at the time of sampling (30 July 2018). In addition, two small streams with contrasting characteristics were
       sampled: i) Y3 (~17 km²), characterised by a relatively high DOC load and low POC load, representing soil leaching and active
       layer drainage (Bröder et al., 2020), and ii) a thaw stream at Duvanny Yar (DY TS; <0.1 km²), characterised by an extremely
       high POC load, and relatively low DOC load, representing eroding Yedoma permafrost (Vonk et al., 2013). At Duvanny Yar,

additional samples were taken from a thawing permafrost headwall, and from the outflow of a thaw stream into the Kolyma
       River (DY KOL) to characterize the Yedoma permafrost endmember and mixing of the thaw streams with Kolyma waters.
       We compare our samples with the data reported in Bröder et al. (2020), including POC samples from the Kolyma River
       (sampled at Cherskiy) and the tributary stream Y3, covering the open-water seasons (late May until late September/early
       October) of 2013 and 2015. These samples were included in the present study to give insight into temporal variations at these

locations, in addition to spatial variations along the transect.
       Furthermore, this new dataset is compared to published data on surface water DOC and POC, and surface sediments from the
       East Siberian Sea. The East Siberian Sea is situated between the Laptev Sea and the New Siberian Islands to the west and the
       Chukchi Sea and Wrangel Island to the east (Fig. 1a). It covers an area of approximately one million square kilometres, and
       has an average depth of 58 meters. Previous publications (Tesi et al., 2014; Vonk et al. 2010, 2012a) have characterized surface

water DOC and POC in the ESS, along with underlying surface sediments, following the paleo river valley of the Kolyma up
       to 600 km offshore (Fig. 1c). The samples along this transect were collected on 3-5 September 2008, and started ca. 12 km
       farther offshore than our farthest river transect point (K6). Data from a more recent cruise (between 31 July and 4 August
       2014) are used to extend this transect up to 1000 km offshore (Bröder et al., 2019; Salvadó et al., 2016). The list of ESS station
       locations and data used in this study can be found in Table A1.

**Table 1. Sample locations, description, short ID, and the distance of each location to the mouth of the Kolyma River.**

| Short ID | Location description | Date sampled | Latitude (°) | Longitude (°) | Distance to ocean (km) |
|---|---|---|---|---|---|
| **Kolyma** | | | | | |
| K 1 | Before Duvanny Yar | 23/07/2018 | N 68.63890 | E 159.12080 | 240 |
| K 2 | At Cherskiy | 31/07/2018 | N 68.77598 | E 161.26494 | 110 |
| K 3 | East branch | 28/07/2018 | N 69.20045 | E 161.44044 | 60 |
| K 4 | West branch | 31/07/2018 | N 69.09501 | E 161.01700 | 60 |
| K 5 | Main delta channel | 28/07/2018 | N 69.53432 | E 161.93555 | 10 |
| K 6 | Outflow to ESS | 28/07/2018 | N 69.67805 | E 162.38632 | 0 |
| **Duvanny Yar** | | | | | |
| DY TS | Yedoma thaw stream | 02/08/2018 | N 68.62987 | E 159.14470 | 230 |
| DY KOL | Kolyma directly at thaw stream outflow | 23/07/2018 | N 68.63060 | E 159.15478 | 230 |
| **Tributaries** | | | | | |
| BA | Bolshoy Anyui | 01/08/2018 | N 68.46015 | E 160.78267 | 160 |
| MA | Maly Anyui | 01/08/2018 | N 68.45193 | E 160.81279 | 160 |
| Y3 | Y3 | 26/07/2018 | N 68.75919 | E 161.44769 | 120 |
| PAN | Panteleikha | 30/07/2018 | N 68.70301 | E 161.51472 | 120 |



### 2.2 Sampling and sample processing

#### 2.2.1 Particulate and dissolved organic matter, and solid phase extractions

About 20 L of surface water was collected in LDPE bags (Vitop, Rink GmbH) in the centre or the fastest flowing part of the
river at each location, except for sample DY KOL, which was sampled at the shore of the Kolyma in the outflow of a thaw
stream (Table 1). Within 12 hours after sampling, the collected surface water was filtered through pre-combusted (400 °C, 12
h including temperature ramping) and pre-weighed glass fibre filters (pore size 0.7 µm, Whatman GF/F). Small GF/F filters
(diameter 47 mm; glass filtration tower, Wheaton) were used for total suspended particulate matter (SPM), POC, and carbon
isotope analyses, whereas large GF/F filters (diameter 90 mm, pore size 0.7 µm, Whatman; custom made, stainless steel
filtration tower) were used to collect larger quantities of suspended material for biomarker analysis. Filters were stored and
transported frozen (-20°C), and freeze-dried before further analyses.

The filtrate (DOC) was stored in pre-combusted 40 mL amber glass vials, acidified to pH 2 with concentrated HCl, and
transported refrigerated (+5 °C) and dark. After subsampling, the remaining filtrate (0.8 to 12.8 L, depending on DOC
concentration) was used for the solid phase extraction (SPE) of DOC, following the method of Louchouarn et al. (2000) and
Spencer et al. (2010). For this purpose, the filtrate was acidified to pH 2 using concentrated HCl (37%) and 2% of methanol
was added to aid extraction efficiency (Spencer et al., 2010). The acidified filtrate was pumped through a pre-rinsed SPE
cartridge (60 mL Mega Bond-Elut C18; Agilent) using a peristaltic pump with flexible silicone tubing (Cole-Parmer instrument
company). The loaded SPE cartridges were stored and transported refrigerated (+5 °C) and dark. Back at the Vrije Universiteit
Amsterdam, the SPE cartridges were extracted by eluting twice with 40 ml of methanol into pre-combusted glass vials, which
were subsequently dried on a hot plate at 40 – 50 °C under a stream of $N_2$. The recovery of the SPE procedure was $63 \pm 7\%$ (n
= 12).

#### 2.2.2 Riverbed sediment organic matter

Riverbed sediments were sampled using a Van Veen grab-sampler, and stored in sterile Whirl-Pak® bags. These samples
represent recently deposited sediment (i.e., with a large fraction of silt and clay) in more quietly flowing locations of the river
and delta. Within 12 hours after collection, sediments were frozen (-20 °C) and remained so during transport. At the laboratory
at the Vrije Universiteit Amsterdam, the samples were freeze-dried, and sieved through a 200 µm and a 63 µm mesh, resulting
in three size fractions of sediment: coarse sand (>200 µm), fine sand (63-200 µm) and a combination of silt and clay (<63 µm).
Particles coarser than silt (>63 µm) are quickly deposited during sediment transport, and carry little mineral-associated OC,
while the fine sediment fraction (<63 µm) carries the bulk of the mineral-associated OC (Coppola et al., 2007; Keil et al.,
1994; Tesi et al., 2016), and is considered to represent an integrated signal of suspended matter transported by the river
(Freymond et al., 2018). Therefore, in this study, we focus only on the fine, easily transportable fraction of the sediment. The
term "SOC" in this paper therefore refers to the OC content of the < 63 µm sediment fraction. This fractionation step allows
us to cross-compare the same fraction of sediment and OC at different locations along the river transect and beyond, on the
shelf, despite the heterogeneity of bulk sediments.

### 2.3 Mineral-specific surface area analysis

For mineral surface area (SA) measurements, subsamples of about 1.5 g freeze-dried sediment were combusted at 450 °C for
12 h to remove OC, rinsed twice with MilliQ to remove salt and ashes, and freeze dried again. Directly prior to analysis, the
samples were degassed for a minimum of 2 hours at 300 °C under vacuum. The analyses were performed at the Vrije
Universiteit Amsterdam on a Quantachrome Nova 4200e, using the 6-point Brunauer–Emmett–Teller method (Brunauer et al.,
1938). The SA measurements were regularly checked against two certified reference materials (5.41 $m^2$ $g^{-1}$ and 27.46 $m^2$ $g^{-1}$).



### 2.4 Bulk elemental analyses

### 2.4.1 Carbon concentrations and stable carbon isotope analyses

Concentration of DOC and DOC-$\delta^{13}$C were analysed with an Aurora1030 TOC analyser coupled to a Delta V Advantage isotope ratio mass spectrometer (IRMS) at KU Leuven (Belgium), see further method details in Deirmendjian et al. (2020).

The POC concentrations, and POC-$\delta^{13}$C were measured on a combined elemental analyser - isotope ratio mass spectrometer (EA-IRMS) at the National Research Council Institute of Polar Sciences (Bologna, Italy). Before subsampling, the concentration of suspended particulate matter (SPM) was determined by weighing the sediment loaded filters after freeze-drying and dividing by the volume of water filtered. Subsamples were punched out of the 47 mm GF/F filters, placed in pre-combusted silver capsules, and weighed. Inorganic C was removed by adding 50 µl of 1 M HCl twice to the silver capsules.

After oven drying (60 °C), the silver capsules were wrapped in tin capsules to aid combustion during analysis.

Sediment (<63 µm fraction) was crushed and homogenized in an agate mortar, and two subsamples of each sample were weighed into pre-combusted silver capsules for total OC and $\delta^{13}$C analyses. The sediment was acidified in as described above for the filters to remove inorganic C, wrapped in tin capsules after acidification and measured for OC at the Sediment Laboratory and for $\delta^{13}$C at the Stable Isotope Laboratory of the Vrije Universiteit Amsterdam (The Netherlands). All $\delta^{13}$C

values are reported in ‰ relative to the VPDB (Vienna Pee Dee Belemnite).

### 2.4.2 Radiocarbon analyses

Radiocarbon ($^{14}$C) analyses were carried out using an EA coupled to a MICADAS accelerator mass spectrometer (AMS) at the Laboratory of Ion Beam Physics of the Swiss Federal Institute of Technology (ETH, Zürich, Switzerland), following the method described in McIntyre et al. (2017). Subsampled filters (POC) and sediment (SOC) were weighed in pre-combusted

silver capsules, and inorganic carbon was removed by fumigation in a desiccator with 37% HCl at 60 °C for 72 h (Komada et al., 2008). After fumigation, samples were dried over NaOH pellets at 60 °C for 72 h to neutralize the acid, and wrapped in tin capsules. The final $^{14}$C results are corrected for constant background contamination using the method described in Haghipour et al. (2018). All radiocarbon data are presented either as $\Delta^{14}$C (‰) or as conventional, uncalibrated radiocarbon age (yr) (Stuiver and Polach, 1977).

### 2.5 Molecular and biomarker analyses

### 2.5.1 CuO oxidation products

Microwave-assisted alkaline CuO oxidation was carried out at the laboratory of the Vrije Universiteit Amsterdam to extract lignin and cutin products from SPE-DOC and SOC samples, following the method of Goñi & Montgomery (2000). In summary, Teflon extraction vessels were loaded with ~2 – 4 mg OC, 500 mg CuO and 50 mg ferrous ammonium sulfate. For

SPE-DOC samples, 10 mg of glucose was added to prevent superoxidation of lignin polymers. Then, 10 mL of degassed 2 N NaOH solution was added under oxygen-free conditions. The oxidation was performed using a MARS 6 microwave (CEM Cooperation) at 150 °C (1,600 W, 8 min ramp, continued heating for 90 min). The resulting extract was centrifuged, transferred to a pre-combusted glass vial, and an internal recovery standard (Ethyl vanillin; Sigma-Aldrich) was added. The samples were acidified to pH 1 by adding concentrated HCl, and then extracted twice with ethyl acetate. The samples were dehydrated with

anhydrous sodium sulfate, transferred to combusted amber glass vials, and dried under flow of N$_2$. Prior to analyses on an Agilent gas chromatograph-mass spectrometer (GC-MS) at the National Research Council Institute of Polar Sciences (Bologna, Italy), samples were re-dissolved in pyridine and methylated with BSTFA. The individual lignin phenols, benzoic acids, and *p*-hydroxybenzenes were quantified by comparison with commercially available standards, and quantification of cutin-derived products was done using the response of trans-cinnamic acid.



**2.5.2 Lipid biomarker analyses**

For the extraction of lipid biomarkers from POC, one to three freeze-dried 90 mm GF/F filters, each containing 3 to 26 mg of POC, were selected per location and placed in pre-extracted Teflon extraction vessels. For riverbed SOC, ~2 g of sediment was weighed in per extraction vessel, containing ~12 – 17 mg of OC. Samples were solvent-extracted twice with 15 ml DCM:MeOH (9:1 v/v) at 100 °C (1,600 W, 5 min ramp, continued heating for 15 min), using a MARS 6 microwave (CEM Cooperation). The resulting extract was saponified with 10 – 15 ml of KOH in methanol (0.5 M) at 70 °C for 2 h. Subsequently, 5 – 10 ml of MilliQ water with 2% NaCl was added. The neutral fraction (*n*-alkanes) was extracted with hexane (3 x 10 ml), after which the samples were acidified to pH 2 with concentrated HCl. The acid fraction was then extracted with hexane:DCM (4:1 v/v), methylated with $BF_3$-MeOH (80 °C, 30 min), and extracted with DCM after addition of MilliQ water. The acid fraction was further cleaned of impurities by column chromatography ($SiO_2$, water-deactivated), by eluting first with hexane, then DCM:hexane (4:1) and DCM. The cleaned methylated *n*-alkanoic acids, concentrated in the DCM:hexane fraction, were then analysed on a GC-MS at the National Research Council Institute of Polar Sciences (Bologna, Italy). Quantification of high molecular weight (HMW; carbon chain length 24 – 30) *n*-alkanoic acids was done by comparison with commercially available standards (alkanoic acid C22, C24, C26, C28 and C30; Sigma-Aldrich). The carbon preference index (CPI) of the HMW *n*-alkanoic acids is calculated as the ratio between even and odd carbon chain lengths (Eq. 1).

$$CPI = \frac{\left(\frac{1}{2}*[C23+C25+C27+C29]\right)+\left(\frac{1}{2}*[C25+C27+C29+C3\ ]\right)}{[C24+C26+C28+C30]}, \tag{1}$$

**2.6 Endmember analyses**

Source apportionment models are commonly used to distinguish different source contributions to the total OC pool based on their isotopic signature. Dual-carbon isotope endmember mixing models have proven to be useful tools to disentangle the various sources of organic matter in different environments, as these Markov chain Monte Carlo (MCMC) techniques account for uncertainties in both the endmember values as well as the uncertainties in sample measurements and thus provide better constraints on the relative contributions of different sources to bulk OC (Andersson et al., 2015; Bosch et al., 2015; Vonk et al., 2012; Wild et al., 2019). For this sample set, we identified three different OC sources that contributed to the POC and SOC, and calculated their relative fractions using a dual-isotope $\delta^{13}$C and $\Delta^{14}$C endmember mixing model. Our approach combines an isotopic mass-balance source apportionment model, Bayesian MCMC, which uses dual-isotope signatures (endmembers) from bulk OC to differentiate between the following three sources: i. Permafrost OC; ii. Modern vegetation and surface soil OC; iii. Riverine primary production OC (for the Kolyma samples) or Marine primary production OC (for the ESS samples). We defined the endmember for permafrost OC as a mixture of Pleistocene Ice complex deposits (ICD) and Holocene permafrost (including Holocene peat), with an $\delta^{13}$C value of -26.3 ± 0.7‰ (Vonk et al., 2012) and a $\Delta^{14}$C value of -761.2 ± 120‰. The $\Delta^{14}$C value was derived as the mean of the ICD endmember (-954 ± 65.8‰; Wild et al., 2019) and the Holocene/peat permafrost endmember (-567 ± 157‰; Wild et al., 2019) assuming approximately equal carbon stock input of these two pools in this region (Zimov et al., 2006a). Different weighing of these two permafrost OC pools (e.g., spatial area-weighing of ICD coverage giving a $\Delta^{14}$C value of -683.7 ± 136‰) did not significantly change the result of the model. The endmember for the second source, modern vegetation and surface soil OC (including the active layer, soil OC and recent vegetation; hereafter "vegetation/soil OC"), was adapted from Wild et al. (2019) with a $\delta^{13}$C value of -27.2 ± 1.1‰ (n = 150) and a $\Delta^{14}$C value of -52.7 ± 137.3‰ (n = 118). Wild et al. (2019) presented endmembers of these sources separately, but we prefer to combine them as one contemporary terrestrial endmember, therefore their values were averaged, equally weighed to one endmember. The third source, primary production OC (fluvial or marine), has an endmember $\delta^{13}$C value of -32.1 ± 3.0‰ and a $\Delta^{14}$C value of +11.0 ± 37 for riverine samples (henceforth named "Riverine PP OC"), while the endmember for marine samples ("Marine PP OC") is $\delta^{13}$C = -24.0 ± 3‰, $\Delta^{14}$C = +60 ± 60‰ (Vonk et al., 2012). The riverine PP OC endmember is based on a compilation of samples and using the endmember values of previous studies: $\delta^{13}$C = -30.5 ± 2.5 ‰, $\Delta^{14}$C = +41.9



± 4.2 ‰ (Winterfeld et al., 2015a), $\delta^{13}C$ = -30.6 ± 3.3‰, $\Delta^{14}C$ = +48 ± 11‰ (Wild et al., 2019), and our own algal bloom sample from the Panteleikha River ($\delta^{13}C$ = -33.5‰, $\Delta^{14}C$ = -26‰). For the marine $\delta^{13}C$ endmember (-24.0 ± 3‰) we also tested a value of -21 ± 1‰, used in Bröder et al. (2016) for ESS sediments. The modelling results showed a minimal, non-significant change for SOC in endmember contributions (Fig. A1). However, for POC, there was a large shift on the first part

of the marine transect from POC being marine PP dominated to being vegetation/soil OC dominated using the -21 ± 1‰ endmember (Fig. A1. This is probably related to the sharp transition from a riverine PP to a marine PP endmember, while in reality the transition is not as sharp and likely a mixture of these two sources within the estuary. The contribution of the permafrost endmember to the POC pool was not significantly affected by this shift. Increasing or decreasing the standard deviation of either of the marine PP endmembers (-24 and -21‰) from ± 1 ‰ to ± 3‰ did not make a difference.

The dual-isotope/three-sources version of the MCMC source apportionment model was adapted from Bosch et al. (2015). We used MATLAB (version 2021a) to model contributions of the three different sources, with the following model parameters: 1,000,000 iterations, a burn-in (initial search phase) of 10,000, and a data thinning of 10. For further details on the method see (Andersson et al., 2015; Andersson, 2011; Bosch et al., 2015).

## 3 Results and Discussion

The Kolyma River transports fluvial organic matter towards the East Siberian Sea in three different compartments; the dissolved, particulate and sedimentary OC pools. Our study targets all these compartments and adds a spatial dimension by not only sampling along a 250-km main stem transect, but also including a range of tributaries, and extending the riverine transect ~1000 km across the ESS using existing data (SI Table 1; Bröder et al., 2019; Salvadó et al., 2016; Tesi et al., 2014; Vonk et al., 2010a). In contrast to previous studies (e.g., Bröder et al., 2020; McClelland et al., 2016), we do not focus on the

seasonal OC variability within fluvial systems (i.e., comparing different stages of the hydrograph), but aim to convey a consolidated picture of riverine dissolved, particulate and sedimentary OC delivered to the East Siberian Sea, and to give insight on the processes that affect these OC pools along the land-ocean continuum.

### 3.1 Three contrasting OC pools: Concentrations of DOC, POC and SOC

In Arctic rivers, DOC and POC concentrations vary significantly during seasons (Bröder et al., 2020; Holmes et al., 2012;

McClelland et al., 2016). Concentrations found in this study match the typical range of DOC and POC values of the Kolyma River in the late summer season (Table 2). The DOC concentrations along the Kolyma River transect range from 2.76 to 4.97 mg L$^{-1}$, which are on the same order of magnitude as DOC in ESS surface waters. The POC concentrations during this period, range from 1.49 to 2.73 mg L$^{-1}$, and show a rapid decrease once offshore in the ESS, from 2.7 mg L$^{-1}$ at location K6 to 0.2 mg L$^{-1}$ approximately 50 km farther at location YS34B (at water depth 10 m; Vonk et al., 2010a, Fig. 1). The Kolyma tributaries

PAN and Y3 show notably higher DOC and POC concentrations of 21.5 and 9.71 mg L$^{-1}$ DOC, and 4.50 and 2.38 mg L$^{-1}$ POC, respectively, compared to the Kolyma. The sample DY TS shows extremely high concentrations of DOC (103 mg L$^{-1}$) and POC (> 7,300 mg L$^{-1}$), which are in the same range as other thaw streams at this location (Vonk et al., 2013). Sample DY KOL, located right at the outflow of a thaw stream into the Kolyma River, shows that the extremely high concentrations of DOC and POC coming from DY thaw streams are quickly diluted by river water and/or settles rapidly to the riverbed. The

DOC concentration in this sample is in the same range as the Kolyma main stem (2.75 mg L$^{-1}$), while the POC concentration remains elevated at 103 mg L$^{-1}$. The SOC concentrations in the <63 μm fraction of riverine sediment show values ranging from 0.45 to 1.0% (average 0.76 ± 0.19%, n = 5) along the Kolyma River transect, which are slightly lower than ESS SOC (which was not sieved) with concentrations between 0.80 and 1.76% (average 1.15 ± 2.94%, n=18; Salvadó et al., 2016; Vonk et al., 2010a). The fraction of OC in particulate matter (OC concentrations normalized to TSS) is much higher than in SOC,



ranging from 6.7 to 12.8% within the Kolyma and with values up to 47% for the Pantaleikha, pointing towards a significant contribution of primary production (i.e., pure organic matter without minerals) to the particulate load.

**Table 2. Bulk data for sediment organic carbon (SOC), dissolved organic carbon (DOC) and particulate organic carbon (POC). Including concentrations, surface area (SA), organic carbon loading and isotopic data δ13C, Δ14C, conventional, uncalibrated radiocarbon age (yr), and fraction modern (Fm) with the measurement error.**

| Short ID | SOC | SA | OC loading | Grain size | | δ13C | Δ14C | 14C age | Fm | Fm error |
|---|---|---|---|---|---|---|---|---|---|---|
| Sediment <63 μm | wt.% | m2 g-1 | mg OC m-2 | median μm | | ‰ | ‰ | yr. | | ± |
| **Kolyma** | | | | | | | | | | |
| K 2 | 0.82 | 9.6 | 0.85 | 39.5 | | -27.1 | -521 | 5850 | 0.483 | 0.003 |
| K 3 | 0.66 | 8.7 | 0.75 | 39.4 | | -26.9 | -586 | 7020 | 0.418 | 0.004 |
| K 4 | 1.00 | 11.9 | 0.84 | 37.0 | | -27.1 | -530 | 6000 | 0.474 | 0.003 |
| K 5 | 0.88 | 11.9 | 0.74 | 31.5 | | -27.4 | -537 | 6120 | 0.467 | 0.005 |
| K 6 | 0.45 | 8.4 | 0.54 | 46.8 | | -27.2 | -579 | 6890 | 0.424 | 0.004 |
| **Duvanny Yar** | | | | | | | | | | |
| DY PF | 0.63 | 13.4 | 0.47 | 34.5 | | -26.1 | -965 | 26800 | 0.036 | 0.001 |
| **DOC and POC** | DOC | DOC δ13C | POC | POC | SPM | POC δ13C | POC Δ14C | POC 14C age | Fm | Fm error |
| | mg L-1 | ‰ | mg L-1 | wt.% | mg L-1 | ‰ | ‰ | yr. | | ± |
| **Kolyma** | | | | | | | | | | |
| K 1 | 2.76 | -28.3 | 1.42 | 12.8 | 11.2 | -31.6 | -221 | 1940 | 0.785 | 0.010 |
| K 2 | 2.96 | -30.0 | 1.71 | 6.68 | 25.6 | -29.0 | -379 | 3760 | 0.627 | 0.007 |
| K 3 | 3.6 | -30.0 | 1.71 | 7.28 | 23.5 | -29.8 | -306 | 2870 | 0.700 | 0.008 |
| K 4 | 3.49 | -29.4 | 1.49 | 8.12 | 18.4 | -30.0 | -362 | 3540 | 0.643 | 0.007 |
| K 5 | 3.23 | -28.8 | 1.67 | 9.10 | 18.4 | -30.7 | -296 | 2750 | 0.710 | 0.010 |
| K 6 | 4.97 | -27.3 | 2.73 | 8.12 | 33.7 | -29.9 | -301 | 2810 | 0.705 | 0.008 |
| **Duvanny Yar** | | | | | | | | | | |
| DY TS | 103 | -29.8 | 7325 | 1.22 | 600363 | -25.5 | -860 | 15730 | 0.141 | 0.002 |
| DY KOL | 2.75 | -28.7 | 15.7 | 4.50 | 348 | -26.3 | -859 | 15670 | 0.142 | 0.003 |
| **Tributaries** | | | | | | | | | | |
| BA | 4.43 | -29.5 | 1.7 | 7.85 | 21.7 | -31.3 | -175 | 1480 | 0.832 | 0.009 |
| MA | 3.16 | -28.9 | 1.29 | 15.9 | 8.11 | -33.3 | -348 | 3370 | 0.658 | 0.006 |
| Y3 | 21.5 | -29.4 | 2.38 | 18.3 | 13.02 | -32.5 | -160 | 1500 | 0.830 | 0.010 |
| PAN | 9.71 | -31.2 | 4.5 | 46.6 | 9.67 | -33.5 | -26 | 145 | 0.984 | 0.010 |


## 3.2 Three contrasting OC pools: Isotopes of DOC, POC and SOC

Each organic carbon pool (DOC, POC and SOC) shows distinctly different stable carbon isotope ($\delta^{13}$C) and radiocarbon isotope ($\Delta^{14}$C) ratios, which are important tools in characterising OC and tracing OC from different sources. The DOC-$\delta^{13}$C along the Kolyma River transect ranges from -27.3 to -30.0‰ (Table 2), which is comparable to earlier published data (Feng

et al., 2013; Mann et al., 2015; Wild et al., 2019). Although we have not measured DOC-$\Delta^{14}$C in this work, earlier studies show that Kolyma River and tributary DOC is relatively young ($\Delta^{14}$C in the range of +150 to -100‰; Neff et al., 2006; Wild et al., 2019). The DOC-$\delta^{13}$C of the tributaries and Duvanny Yar are in the same range as the Kolyma, except for sample PAN, which shows a more depleted $\delta^{13}$C value of -31.2‰. An earlier study on a Duvanny Yar thaw stream found DOC-$\Delta^{14}$C values between -974 and -911 ‰ (up to 30,000 yr old) (Vonk et al., 2013). However, such old aged DOC has not been found in the

main Kolyma River, likely due to rapid turnover times of permafrost DOC in Arctic waters (Rogers et al., 2021).

The $\delta^{13}$C of Kolyma POC ranges from -29.0 to -31.6‰, and the $\Delta^{14}$C ranges from -221 to -379‰, corresponding to 1,940 to 3,760 yr (Fig. 2). The $\delta^{13}$C values of the Kolyma transect correlate with POC% (the OC weight % of dried particulate matter;



R$^2$ = 0.91, p < 0.01) and Δ$^{14}$C (R$^2$ = 0.79, p < 0.01), in other words, samples with a high POC% have a more depleted δ$^{13}$C

value and a less depleted (younger) Δ$^{14}$C value, both supporting a significant contribution from riverine production. The

tributary and Duvanny Yar samples are clearly different from the Kolyma in their isotopic signature of POC: the two Duvanny

Yar POC samples are more enriched in δ$^{13}$C (-25.5 and -26.3‰), and substantially more Δ$^{14}$C depleted (-859 and -860‰;

15,700 yr) than Kolyma POC, while the other tributaries (PAN, MA, BA and Y3) show generally more depleted δ$^{13}$C values

(-31.3 to -33.5‰) and younger Δ$^{14}$C values (-26 to -348‰; 145 to 3,370 yr) than Kolyma POC. The Δ$^{14}$C-POC values in the

Kolyma (K1-K6) are in the same range as Kolyma summer POC of 2013 and 2015 (-314 ± 83‰, n = 38; Bröder et al., 2020;

Fig. 2), and slightly younger than the average Kolyma summer POC between 2003 and 2011 (-463 ± 15‰, n = 32; McClelland

et al., 2016).

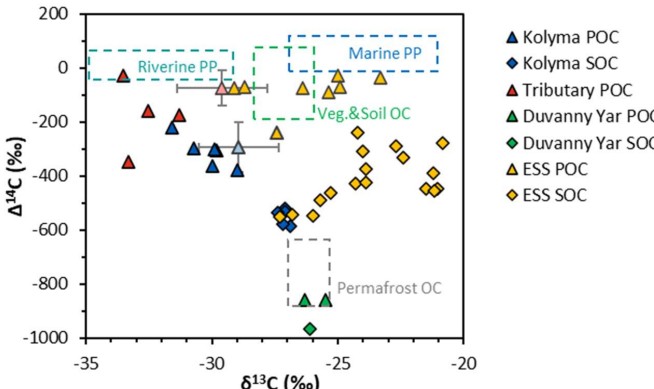

**Figure 2. Δ$^{14}$C versus δ$^{13}$C of the Duvanny Yar (DY), Kolyma, tributary, and East Siberian Sea samples. Triangles are POC samples, diamonds are SOC samples. Kolyma main stem in dark blue, ESS in yellow, tributaries in red and DY in green. The boxes represent the endmembers as defined in section 2.6. The triangles with standard deviation show the average of Kolyma particulate organic**
**carbon (POC, faded blue) and Y3, one of the tributaries, POC (faded red) samples of Bröder et al. (2020) for reference.**

These trends in δ$^{13}$C and Δ$^{14}$C in POC point towards the influence of a younger, more $^{13}$C-depleted source of OC in the Kolyma

River and especially in the tributaries. A similar trend was found in Bröder et al. (2020), suggesting influence of riverine

primary production. In situ production of OC by fluvial organisms in Arctic rivers and streams has not received a lot of

attention, but frequently displays depleted δ$^{13}$C signatures (e.g., -30.5 ± 2.5‰ in Winterfeld et al. (2015a) Lena River; -30.6 ±

3.3‰ Ob and Yenisey rivers, Galimov et al. (2006); -33.4 ± 4.2‰ in Shakil et al. (2020), streams on the Peel Plateau). The

lower δ$^{13}$C values of heterotrophic OC are due to contributions of recycled $CO_2$ that sources from terrestrial organic matter

breakdown, which is already relatively depleted in δ$^{13}$C (Meyers, 1994). Winterfeld et al. (2015a) applied a source-

apportionment approach to Lena River POC in summer and found that primary production accounted for up to 80% of the

fluvial POC. This 'recycled carbon' (Wild et al., 2019) appears to be an important component of summer POC transport, which

is reflected in the overall δ$^{13}$C-depleted values in the Kolyma River and tributaries POC pool (Fig. 2, and section 3.3).

Comparing the carbon isotope data of fluvial POC collected in this study with surface water POC collected along the extended

Kolyma River transect in the ESS shows a large difference between the terrestrial and marine samples (Fig. 2; Fig. 3b and d).

The ESS POC is distinctly younger than Kolyma POC: Δ$^{14}$C values between -28 and -75‰ for the inner ESS and between -

69 and -240‰ for the outer ESS, and similarly more enriched in δ$^{13}$C: ranging from -23.3 to -29.1‰, with a trend towards

more enriched values moving from the river mouth farther offshore (Salvadó et al., 2016; Vonk et al., 2010a).

We find that Kolyma SOC is distinctly older (Δ$^{14}$C of -521 to -586‰; 5,850 to 7,020 yr) and less depleted in δ$^{13}$C than POC,

displaying a narrow range in δ$^{13}$C values (-26.9 to -27.4‰) (Fig. 2; Fig 3a, c). On the other hand, Kolyma River SOC is

distinctly younger than the Yedoma permafrost material from Duvanny Yar (Fig. 2). The Yedoma permafrost sample DY PF

shows an extremely depleted Δ$^{14}$C value of -965‰ (26,800 yr), and a slightly less depleted δ$^{13}$C value of -26.1‰. The ESS

SOC close to shore shows a similar age as the Kolyma SOC in this study, with a trend towards less depleted Δ$^{14}$C values





farther offshore (-624 to -332‰, Δ14C). In ESS SOC-δ13C, a trend can be seen moving from -27.1 ‰, close to the value of
Kolyma SOC, towards more enriched δ13C values of -22.4‰ farther offshore (Fig. 3a). This increase in δ13C and Δ14C values
of POC and SOC moving from river to shelf is likely due to the increased contribution of marine PP OC to the SOC and POC
pools farther offshore, together with sorting and settling of terrestrial and permafrost-derived OC (Bröder et al., 2018; Tesi et
al., 2014; Vonk et al., 2012), processes that will be discussed in more detail in the next sections.

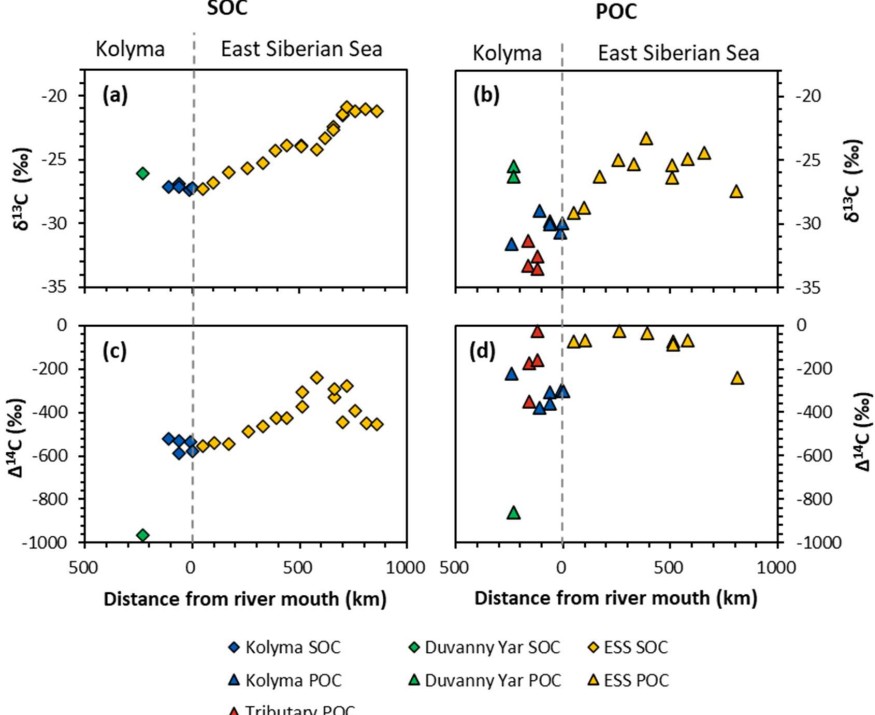

**Figure 3. Carbon isotopes over transect distance, with the riverine part on the left side of each figure, and the marine part on the
right. (a) δ13C ratio (in ‰ relative to VPDB) of Duvanny Yar (DY; green), Kolyma (blue) and East Siberian Sea (ESS; yellow)
sediment organic carbon (SOC). (b) δ13C ratio of DY (green), Kolyma (blue), tributaries (red), and ESS (yellow) particulate organic
carbon (POC). (c) Δ14C ratio of DY, Kolyma and ESS SOC. The lower the Δ14C ratio, the older the material is. (d) Δ14C ratio of DY,
Kolyma, tributaries and ESS POC.**

### 3.3 Quantifying the sources of OC: End member mixing analyses

The Δ14C and δ13C signatures of POC and SOC can be used to quantify the relative contributions of different organic carbon
sources (i.e., permafrost OC; vegetation/soil OC; riverine PP OC for the Kolyma and marine PP OC for the ESS) to these two
carbon pools, following the method described in section 2.6. We revisit the endmember mixing results from Vonk et al., (2012)
for the ESS and from Wild et al., (2019) for the Kolyma River to connect river and shelf environments with the newly defined
endmembers (see section 2.6 for endmember definitions). Relative contributions of the different sources varied considerably
between POC and SOC, and Kolyma and ESS (Fig. 4).

Along the river transect, Kolyma River main stem POC consists largely of riverine PP OC (44 ± 10%), while the tributary
POC shows even higher values of 64 ± 10% on average. Particularly the Pantaleikha POC stands out with a 79 ± 11% riverine
PP OC contribution, which agrees with our field observation of an algal bloom at the time of sampling. The contribution of
vegetation/soil OC is roughly equal for POC and SOC, ranging from on average 18 ± 14% in Kolyma main stem POC, 15 ±
11% in tributary POC and 19 ± 12% in Kolyma SOC. Permafrost OC is the dominant source in Kolyma SOC (on average 68
± 10%), and the second largest contributor to the Kolyma main stem POC (38 ± 8%). As expected, the contribution of
permafrost OC is highest in Duvanny Yar POC and SOC samples (93 ± 4%).





Source apportionment modelling on the Kolyma POC data from Bröder et al. (2020) shows that the average contribution of permafrost OC to Kolyma POC is in the same range (38 ± 9%) as in this study over their whole sampling period (ranging from Spring to Fall of 2012 to 2015), while the average contribution of vegetation/soil OC is slightly (26 ± 16%) higher, and the average contribution of riverine PP OC is slightly (37 ± 12%) lower than in the samples of this study. This could be due to the timing of the sampling; Bröder et al. (2020) also include the early and late summer when riverine PP may not be high, while

our study likely includes the peak of the riverine PP production. At tributary Y3, including this dataset, the average contribution of permafrost OC is only 11 ± 6%. The bulk of the POC in Y3 comes from the other two sources: 51 ± 19% from riverine and 38 ± 20% from vegetation/soil, which is in line with the conclusion of Bröder et al. (2020), that the Y3 tributary does not have the power to erode deeper into the permafrost.

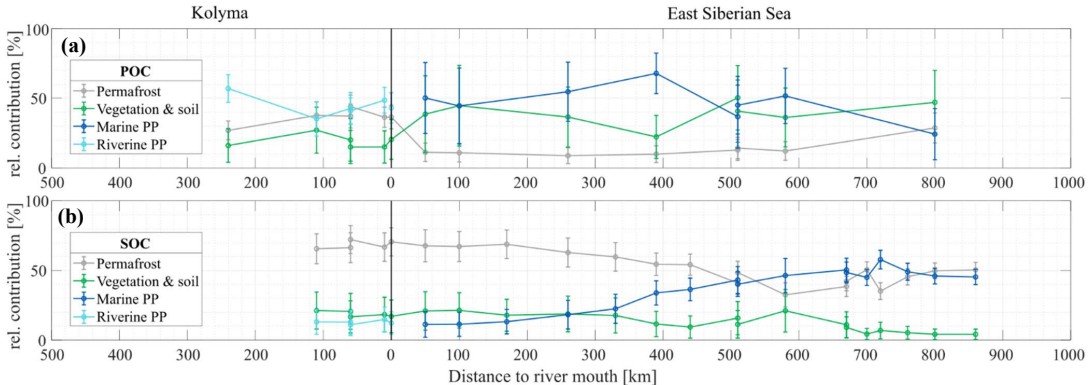

**Figure 4. Relative contribution of three endmembers over transect distance for (a) surface water particulate organic carbon (POC) and (b) sediment organic carbon (SOC), based on dual carbon isotope ($\Delta^{14}$C and $\delta^{13}$C) endmember analyses (EMMA). For the riverine part of the transect (Kolyma; left side of the figure), the endmembers are: Permafrost organic carbon (OC) in grey, Vegetation/soil OC in green, and Riverine primary production (PP) OC in cyan. For the marine part of the transect (East Siberian Sea; right side of the figure), the endmembers are: Permafrost OC in grey, Vegetation/soil OC in green, and Marine primary**

**production OC in dark blue. Definition of the endmembers are described in section 2.6 and can be seen Fig. 2.**

     For the marine transect, the riverine primary production endmember was 'replaced' with the marine primary production endmember, since marine primary production is absent in the river, and riverine primary production OC is thought to be rapidly recycled in a marine setting, which is supported by the rapid shift towards a less depleted $\delta^{13}$C signature of POC in the first part of the offshore ESS transect. Marine primary production appears to be the dominant source of POC in the ESS, supplying

roughly half of the OC along the entire ESS transect (average 47 ± 12%). Furthermore, we find similar results as Vonk et al., (2012; 2010) for ESS SOC: an increase in the contribution of marine PP OC (from 10% to ~50%; Fig. 4b), and a steady decrease of the two terrestrial endmembers farther offshore. Notably, the permafrost OC endmember remains the dominant source of OC up to 500 km offshore, decreasing from 70% to ~40%, before marine PP OC becomes dominant. Note that we have not incorporated lateral transport times of sediment OC (estimated up to 3600 yr. across the Laptev shelf; Bröder et al.,

2018) that affect all terrestrial OC during sedimentary transport. In contrast, for POC we find an initial sharp decrease in the contribution of permafrost OC, from ~40% in the Kolyma River to ~10% at the first transect point offshore (Fig. 4a), remaining around 10% for the entire length of the transect. Likely, the permafrost OC consists mostly of mineral-bound OC, material that has been shown to rapidly settle in the near-shore region (Jong et al., 2020; Karlsson et al., 2011; Vonk et al., 2010b), which may explain the high permafrost contribution to the SOC pool and rapid decrease of permafrost OC in the POC pool. On the

contrary, primary production biomass and organic debris is not mineral-bound and can remain afloat over long distances (Karlsson et al., 2011; Vonk et al., 2010b), which is reflected in the relatively high contributions of these pools in the POC of the ESS.



**3.4 Sources of OC: lignin biomarker concentrations and proxies**

Terrigenous biomarkers such as lignin derived phenols, cutin-derived hydroxy fatty acids and HMW *n*-alkanoic acids can be
used to further trace the source, pathway, and fate of OC in rivers and in the marine environment (e.g., Freymond et al., 2018; Tesi et al., 2014). The lignin content, either normalised to OC content (as mg g OC$^{-1}$) or to mineral surface area (as μg m$^{-2}$), refers to the sum of vanillyl (V), syringyl (S) and cinnamyl (C) phenols, and is an indicator for the contribution of higher vascular plant material to the total organic matter pool (Goñi & Hedges, 1992). The ratios between lignin phenol groups S/V and C/V can be used for tracing the various types of plants generating these phenols (Hedges and Mann, 1979). These lignin
source proxies have been extensively used to characterize and trace different pools of OC on land, in rivers and in the marine environment (e.g., Amon et al., 2012; Goñi et al., 2000).

The lignin concentrations in DOC of the Kolyma transect range from 1.70 to 5.11 mg g OC$^{-1}$ (Table 3), and the DOC lignin concentration in the four tributaries (MA, PAN, Y3 and BA) are in the same range as the Kolyma transect (3.61 to 5.62 mg gOC$^{-1}$). These are in the same range as earlier results in the Kolyma River (4.7 mg g$^{-1}$ OC; Feng et al., 2017; 6.5 mg g$^{-1}$ OC
Amon et al., 2012). Sample DY TS is a notable exception, with a higher lignin concentration of 11.79 mg gOC$^{-1}$. For the ESS, only a few dissolved lignin concentrations are published, and they are an order of magnitude lower than in the Kolyma River, at roughly 0.2 mg gOC$^{-1}$ at four points in the outer ESS (Salvadó et al., 2016). Lignin concentrations in Kolyma SOC (< 63 um) range from 6.46 to 14.87 mg gOC$^{-1}$, which is clearly higher than in DOC, but lower than in the first two sampling points in the ESS (not sieved; 28.40 and 16.00 mg gOC$^{-1}$; Salvadó et al., 2016). Farther offshore, the lignin concentrations in SOC
gradually decrease to 0.10 mg gOC$^{-1}$, indicating a decreasing influence of terrestrial biomass on the total OC pool, which is supported by the bulk C isotopes (Salvadó et al., 2016).

A similar pattern is evident in cutin concentrations. For the Kolyma transect, they range from 0.40 to 1.35 mg gOC$^{-1}$ for DOC and 2.18 to 5.67 mg gOC$^{-1}$ for SOC. The cutin concentrations of the four tributaries are in the same range (0.52 to 1.23 mg gOC$^{-1}$) as for Kolyma DOC, while in the Duvanny Yar thaw stream (DY TS) the concentrations are much higher at 4.79 mg
gOC$^{-1}$. The cutin to lignin ratio is higher for SOC than for DOC in the main Kolyma samples (0.40 ± 0.12 versus 0.21 ± 0.11, respectively; Fig. A2), which could be due to a methodological bias: the SOC cutin to lignin ratio could be artificially raised by the sieving step while processing the sediments, thus the lignin-rich organic debris could remain in the coarse fraction (Tesi et al., 2016).

To further pinpoint the source of higher plant-derived OC, ratios between lignin phenol groups (C/V and S/V) can be used to
distinguish different vegetation sources: woody versus non-woody material and gymnosperm versus angiosperm material (Hedges & Mann, 1979; Goñi & Hedges, 1992; Goñi & Montgomery, 2000). The S/V and C/V ratios show fairly consistent values for the Kolyma transect DOC, with an S/V ratio of 0.41 to 0.48 and a C/V ratio of 0.12 to 0.18 (Fig. 5). For the Kolyma SOC, the S/V and C/V ratios are slightly higher than those for DOC (0.51 to 0.54 and 0.21 to 0.48). These ratios indicate a roughly equal mix of gymnosperm and angiosperm material in both DOC and SOC within the Kolyma main stem (Fig. 5), and
an equal mix of woody and non-woody material in DOC. This is in accordance with earlier studies on the sources of DOC in the Kolyma River (Amon et al., 2012), and indicative of the mixed vegetation of the Kolyma watershed (taiga- and tundra vegetation). The DOC sample DY TS appears to consists completely of non-woody angiosperm organic matter, with very high S/V and C/V ratios of 0.97 and 0.31, respectively. However, the DOC sample at DY KOL and the permafrost SOC sample DY PF show to be more of a mix of angiosperm and gymnosperm soft tissue material (S/V 0.55, C/V 0.33), in the same range
as Kolyma SOC and DOC, and in line with other studies on Yedoma deposits (Winterfeld et al., 2015b; with 0.51 – 1.24 for S/V and 0.27 – 1.07 for C/V, Lena Delta Pleistocene and Holocene deposits).



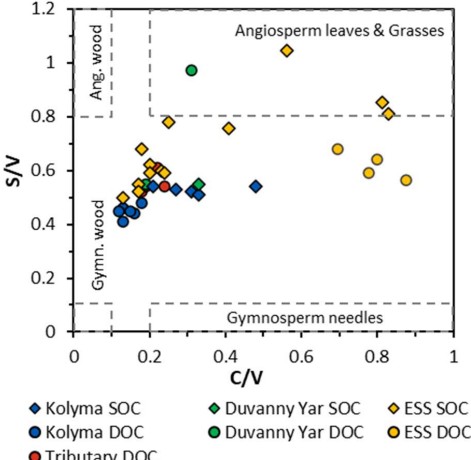

**Figure 5. The ratios between syringyl/vanillyl (S/V) and cinnamyl/vanillyl (C/V) can be used as biomarker source proxies for sediment organic carbon (SOC; diamonds), dissolved organic carbon (DOC; circles) and particulate organic carbon (POC; triangles) from the Kolyma River (blue), Duvanny Yar (green), East Siberian Sea (ESS; yellow) and a couple smaller tributaries of the Kolyma (red). Vegetation boxes are based on (Goñi et al., 2000).**

### 3.5 Sorting and degradation of OC along the land-ocean continuum: OC and biomarker loading

Changes in surface area normalized concentrations (i.e., loadings) of terrestrial OC at bulk and molecular level, measured across coastal shelves (e.g., Tesi et al., 2016) and along riverine transects (e.g., Freymond et al., 2018), provide means to quantify loss of OC due to degradation (Aller & Blair, 2006; Keil et al., 1997). Due to the large variability in hydrodynamic conditions and heterogeneous sediments found within a river, and even more so along a river to shelf transect, it is necessary to trace a specific fraction (e.g. through a consistent method of sediment fractionation) to be able to directly compare sediments across dynamic land-ocean transects. Mineral SA normalization is useful for the fraction high in mineral-bound OC (generally the <63 µm fraction: silt and finer), but works less well on sediment with either large fractions of non-mineral bound carbon (e.g., loose organic debris) or material low in mineral-bound OC (e.g., coarse sand). In addition, the <63 µm fraction is the most easily transported fraction of sediment in rivers, even at lower flow velocities, and is thus the fraction that is transported farthest offshore, as coarse organic debris and sand quickly settles near the coasts (Tesi et al., 2016; Wakeham et al., 2009).

The SA-normalised OC-loadings of the <63 µm river sediment range from 0.54 to 0.85 mg OC m$^{-2}$, with no apparent trend along the main stem transect. These OC loadings are within the range of "typical" river-influenced sediments (0.4 – 1.0 mg OC m$^{-2}$; (Keil et al., 1994; Mayer, 1994; Blair & Aller, 2012), and similar to OC loadings found in other river systems (Freymond et al. (2018); Danube River; similar sediment sampling protocol) and are on average higher than for the surface sediments of the ESS (Bröder et al., 2019; 0.19 to 0.46 mg OC m$^{-2}$) that also show a decreasing trend in OC loadings with increasing distance from land/water depth, suggesting that loss of mineral-bound OC occurs during offshore transport (Keil et al., 1997).

For biomarkers, the SA-normalized lignin concentrations of Kolyma River sediment vary between 5.13 and 12.67 µg m$^{-2}$ (Fig. 6a), while the SA-normalized cutin concentrations are lower, ranging between 1.17 and 4.18 µg m$^{-2}$ (Fig. 6b). The SA-normalised HMW acid concentrations are between 2.26 and 2.92 µg m$^{-2}$ (Fig. 6c). For Duvanny Yar, the OC and biomarker loadings are lower than for Kolyma sediments, due to the combination of high SA and lower OC and biomarker concentration. Freymond et al. (2018) found HMW *n*-alkanoic acid loadings of 0.4-1.5 µg m$^{-2}$, and lignin loadings between 0.6 and 26.4 µg m$^{-2}$ in sediments from the Danube river and its tributaries, using the same sampling and similar extraction methods, which is on the same order of magnitude as for the Kolyma River.



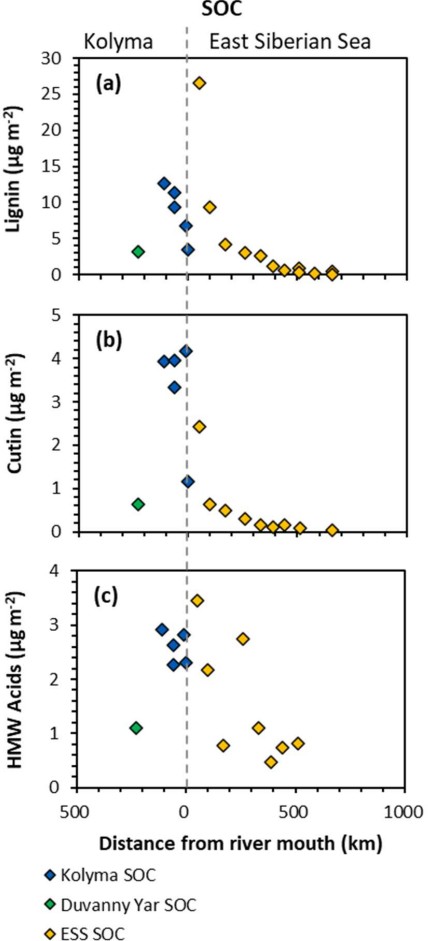

**Figure 6. Biomarker concentrations normalised to mineral surface area over transect distance for the Kolyma (blue), Duvanny Yar (green), and East Siberian Sea (ESS; yellow) sediment organic carbon (SOC) samples. (a) Lignin concentrations (µg m$^{-2}$), (b) Cutin concentrations (µg m$^{-2}$), and (c) HMW *n*-alkanioc acids (C24 – C30) concentrations (µg m$^{-2}$).**

Along the ESS transect sediments, a sharply decreasing trend in lignin loadings was found, from 28.4 µg m$^{-2}$ near the coast down to 0.1 µg m$^{-2}$ on the outer shelf (Fig. 6a), and a similar trend for cutin, from 2.4 to 0.1 µg m$^{-2}$ (Fig. 5b), and HMW acids from 3.4 to 0.8 µg m$^{-2}$ (Fig. 6c) (Karlsson et al., 2011; Tesi et al., 2014; Vonk et al., 2010a). When comparing the marine end of the riverine transect with the shallowest sample of the marine transect, the lignin and cutin concentrations seem disconnected at this riverine-marine interphase: the lignin loadings of the riverine sediment samples appear lower than expected while the

cutin concentrations appear higher than expected, which is also reflected in the cutin/lignin ratios (Fig. A2). Only the riverine HMW acid concentrations align well with the beginning of the marine transect. This discrepancy in behaviour of different biomarkers could be due to their different affinity towards mineral particles. We recognise that we are comparing sieved (<63 µm) riverbed samples with non-sieved (bulk) marine samples but as Tesi et al., (2016) showed that 88 – 95% of the marine sediments in the eastern part of the ESS consist of the <63 µm fraction, we do not think that this size difference is the main

contributor to the observed discrepancy. Instead, earlier sediment partitioning studies that fractionated sediments by both density and size, found that lignin is mostly present in the low-density fraction (<1.8 g mL$^{-1}$) as coarse organic debris (Wakeham et al., 2009; Tesi et al., 2016). It is true that low-density material is often relatively large in size, which in our case (sieving river sediments through 63 µm) places low density material in the coarse fraction. This may explain the lower





concentrations of lignin for the riverine transect. In contrast, cutin-derived acids are more closely associated to fine mineral particles, and HWM acids are more evenly distributed among sediment fractions (Tesi et al., 2016), explaining the better match of these two biomarker groups in comparing Kolyma <63 μm sediment to the ESS transect.

Freymond et al. (2018) propose normalizing OC and biomarkers to SA as the benchmark for comparing river and marine sediments. However, our results point out that this approach seems to work only for certain biomarker groups, since the method we apply based on Freymond et al. (2018), appears to underestimate lignin and overestimate cutin concentrations. Therefore,

we propose to use sediment fractionation methods not purely on size but also on density, and to apply these techniques consistently for all samples, ideally along transects that stretch across the entire river – shelf continuum. While multiple fractionation steps are often time and labour intensive, our results suggest that fractionating only by size (i.e., sieving over 63 μm) is not enough to completely resolve sorting and degradation dynamics of terrestrial OC across the dynamic land-ocean interface, since certain biomarker groups have affinity for different (density) fractions of the sediment.

## 495 3.6 Degradation state of OC along the land-ocean continuum: biomarker proxies

The relative abundances of specific lignin phenol compound classes can be used as proxies for the overall degradation status of organic carbon, for instance, the acid/aldehyde ratios of vanillyl (Vd/Vl) and syringyl (Sd/Sl) phenols are often used as an indicator for degradation of plant organic matter (Hedges et al., 1988; Opsahl & Benner, 1995). The aldehydes degrade faster than the corresponding acids, meaning that a higher Vd/Vl or Sd/Sl ratio indicates more degraded material. However, these

ratios are also influenced by leaching and adsorption processes (Hernes et al., 2007). Another CuO-oxidation product that is frequently used as a degradation indicator is 3,5-dihydroxybenzoic acid (3,5Bd), due to the recalcitrant nature of 3,5Bd, the ratio 3,5Bd/V increases with OC degradation in soils and sediments (Houel et al., 2006). In addition to CuO-oxidation products, the ratio between odd and even HMW $n$-alkanoic acids, the carbon preference index (CPI), can be used as a degradation proxy. The CPI is indicative of organic matter maturity, since fresh plant material has a strong even-over-odd preference for $n$-alkanoic

acids (Freeman and Pancost, 2014; Eglinton and Hamilton, 1967), which is lost with ongoing degradation Thus, organic matter with lower CPI values is considered to be more degraded.

SOC and DOC in the Kolyma River display distinctly different Vd/Vl and Sd/Sl ratios. The SOC shows low values of Vd/Vl and Sd/Sl, ranging from 0.21 to 0.48 and 0.39 to 0.47, respectively (Fig. 7a, c), indicating that SOC is relatively fresh and not degraded. In contrast, the DOC shows higher values, indicating more degradation , with Vd/Vl ranging from 1.77 to 2.35, and

Sd/Sl ranging from 1.06 to 1.39 (Fig. 7b, d). Similarly, the 3.5Bd/V ratios are lower for Kolyma SOC (0.07 – 0.14) than for DOC (0.20 to 0.32) (Fig. 7e. f). In the tributaries, a wider range of Sd/Sl and Vd/Vl ratios was found in DOC, ranging from 0.67 to 3.83 and from 1.22 to 4.89, respectively. Notably, the Yedoma thaw stream (DY TS) shows the highest Vd/Vl and Sd/Sl ratio, and, in contrast, the lowest 3,5Bd/V ratio among all DOC samples, meaning it is fresh in terms of 3,5Bd/V, but degraded in terms of Vd/Vl and Sd/Sl. The CPI of HMW $n$-alkanoic acids, measured in SOC and POC, is slightly lower (i.e.,

more degraded) in SOC (5.20 to 6.45) than in POC (6.32 to 7.63) in the Kolyma River, while sample DY PF shows a higher (i.e., fresher) CPI of 7.65 (Fig. 7g, h). In an earlier study on Yedoma permafrost (or "ice complex deposits") in the Lena Delta (Sánchez-García et al., 2014), a wide range of CPI values was found, between 3.0 and 12.0. The average CPI of HMW $n$-alkanoic acids of Sánchez-García et al. (2014) is however very close to our sample DY PF (6.6 ± 2.7, n = 17).  For two POC samples (K1 and K4), odd HMW $n$-alkanoic acid concentrations were below the detection limit, so the CPI of these two

samples could not be calculated.




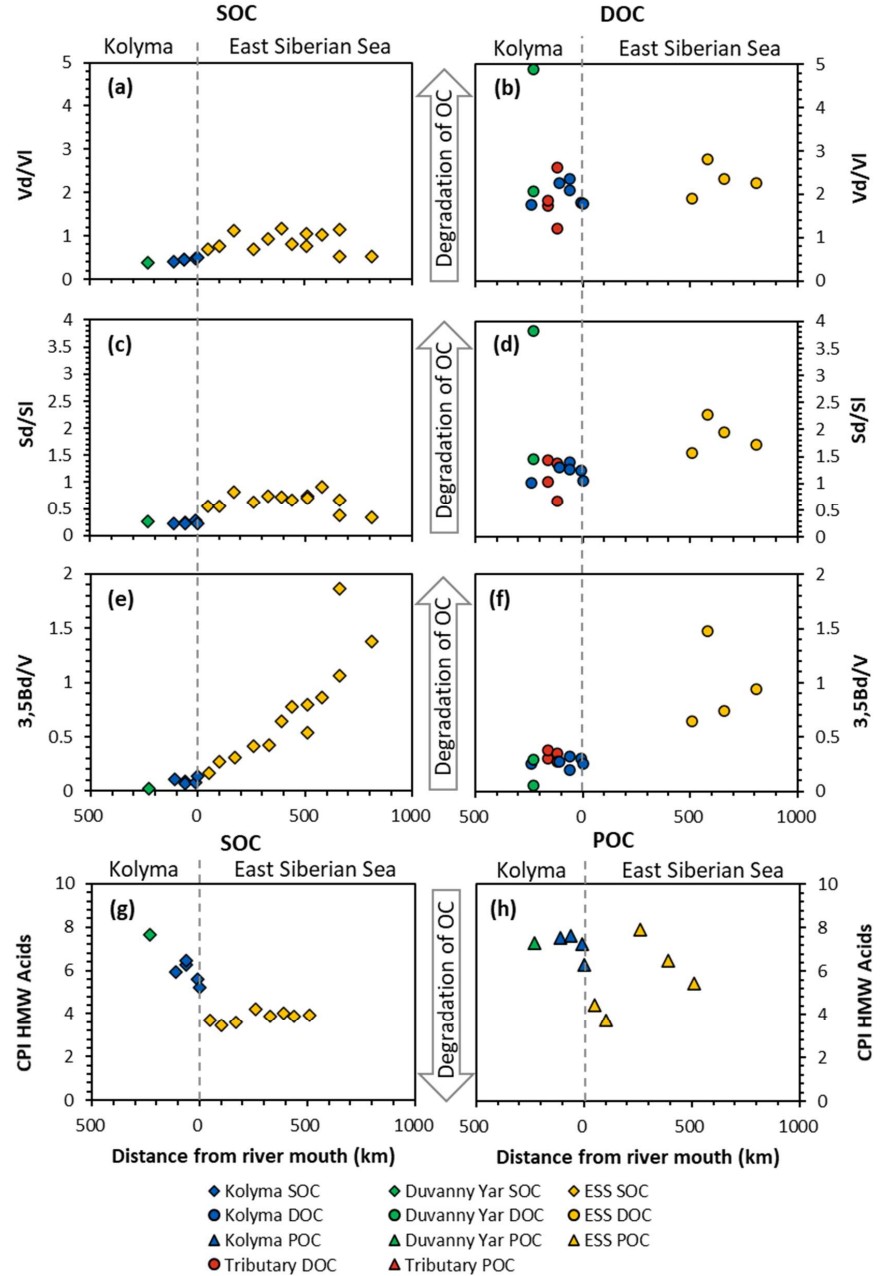

**Figure 7. Biomarker degradation proxies of sediment organic carbon (SOC; diamonds), dissolved organic carbon (DOC; circles) and particulate organic carbon (POC; triangles) from the Kolyma River (blue), Duvanny Yar (green), East Siberian Sea (ESS; yellow) and a couple smaller tributaries of the Kolyma (red) over transect distance. For visual aid the arrows in the middle point towards more degraded values. (a) Acid/aldehyde ratio of Vanillyl phenols (Vd/Vl) of SOC, and (b) of DOC. (c) Acid to aldehyde ratio of Syringyl phenols (Sd/Sl) of SOC, and (d) of DOC. (e) 3,5Bd/V ratio of SOC, and (f) DOC. (g) the carbon preference index of HMW *n*-alkanoic acids (C24 – C30) of SOC, and (h) of POC.**



**Table 3.** Molecular biomarker data for sediment organic carbon (SOC), dissolved organic carbon (DOC), and particulate organic carbon (POC).

| Short ID | Lignin Λ (mg gOC$^{-1}$) | Lignin Λ (µg/m2) | Cutin (mg gOC$^{-1}$) | Cutin (µg m$^{-2}$) | Cutin/Lignin | S/V | C/V | Sd/Sl | Vd/Vl | 3,5-Bd/V | HMW acids* (µg gOC$^{-1}$) | HMW acids* (µg m$^{-2}$) | CPI |
|---|---|---|---|---|---|---|---|---|---|---|---|---|---|
| **Kolyma** | | | | | | | | | | | | | |
| **Sediment <63 µm** | | | | | | | | | | | | | |
| K 2 | 14.87 | 12.67 | 4.62 | 3.94 | 0.31 | 0.53 | 0.27 | 0.23 | 0.41 | 0.11 | 3430 | 2.92 | 5.94 |
| K 3 | 12.46 | 9.39 | 5.26 | 3.96 | 0.42 | 0.52 | 0.31 | 0.25 | 0.45 | 0.09 | 3002 | 2.26 | 6.27 |
| K 4 | 13.43 | 11.34 | 3.95 | 3.34 | 0.29 | 0.51 | 0.33 | 0.22 | 0.45 | 0.07 | 3114 | 2.63 | 6.45 |
| K 5 | 9.19 | 6.77 | 5.67 | 4.18 | 0.62 | 0.54 | 0.48 | 0.29 | 0.47 | 0.08 | 3836 | 2.83 | 5.60 |
| K 6 | 6.46 | 3.46 | 2.18 | 1.17 | 0.34 | 0.54 | 0.21 | 0.22 | 0.49 | 0.14 | 4309 | 2.31 | 5.20 |
| **Duvanny Yar** | | | | | | | | | | | | | |
| DY PF | 6.75 | 3.15 | 1.36 | 0.64 | 0.20 | 0.55 | 0.33 | 0.27 | 0.39 | 0.02 | 2344 | 1.09 | 7.65 |
| **DOC** | | | | | | | | | | | | **POC** | |
| **Kolyma** | | | | | | | | | | | | | |
| K 1 | 3.11 | | 1.35 | | 0.43 | 0.44 | 0.16 | 1.02 | 1.77 | 0.25 | 27^ | | |
| K 2 | 4.24 | | 0.96 | | 0.23 | 0.41 | 0.13 | 1.31 | 2.27 | 0.27 | 798 | | 7.56 |
| K 3 | 4.24 | | 0.66 | | 0.16 | 0.46 | 0.13 | 1.39 | 2.35 | 0.32 | 271 | | 7.63 |
| K 4 | 5.11 | | 0.59 | | 0.12 | 0.45 | 0.15 | 1.27 | 2.1 | 0.2 | 99^ | | |
| K 5 | 1.7 | | 0.40 | | 0.24 | 0.45 | 0.12 | 1.24 | 1.8 | 0.3 | 254 | | 7.25 |
| K 6 | 5.08 | | 0.44 | | 0.09 | 0.48 | 0.18 | 1.06 | 1.78 | 0.25 | 368 | | 6.32 |
| **Duvanny Yar** | | | | | | | | | | | | | |
| DY TS | 11.79 | | 4.79 | | 0.41 | 0.97 | 0.31 | 3.83 | 4.89 | 0.05 | | | |
| DY KOL | 5.31 | | 1.27 | | 0.24 | 0.55 | 0.19 | 1.45 | 2.06 | 0.29 | 898 | | 7.3 |
| **Tributaries** | | | | | | | | | | | | | |
| BA | 5.62 | | 0.52 | | 0.09 | 0.52 | 0.18 | 1.03 | 1.74 | 0.3 | | | |
| MA | 3.61 | | 1.23 | | 0.34 | 0.61 | 0.22 | 1.43 | 1.85 | 0.38 | | | |
| Y3 | 4.6 | | 0.54 | | 0.12 | 0.54 | 0.19 | 0.67 | 1.22 | 0.35 | | | |
| PAN | 4.99 | | 0.61 | | 0.12 | 0.54 | 0.24 | 1.37 | 2.61 | 0.27 | | | |

\* HMW *n*-alkanoic acids with chain lengths C24 – C30

^ Not enough to calculate CPI



When connecting the Kolyma transect to the ESS, both the Vd/Vl, Sd/Sl and 3,5Bd/V ratios are consistently higher in ESS
        sediments than in Kolyma sediments (Fig. 7a, c, e). While Vd/Vl and Sd/Sl show no clear trend across the entire transect, the
        trend in 3,5Bd/V appears to connect well with the riverine transect, with increasing ratios (i.e., more degraded OC) farther
        offshore. For DOC, data from the outer ESS show similar Vd/Vl ratios, and slightly higher Sd/Sl and 3,5Bd/V ratios than
        riverine DOC (Fig. 7b, d, f). The CPI of ESS SOC clusters around ~4 (Fig. 7g), which is considerably lower (i.e., more

degraded) than Kolyma SOC.  For POC, the CPI does not show a clear trend across the river-shelf transect (Fig. 7h), however,
        it remains higher (i.e., fresher) than SOC, which is in line with the results of Salvadó et al., (2016).
        Our results for all four degradation proxies (Vd/Vl, Sd/Sl, 3.5Bd/V, and CPI acids) in the sedimentary OC pool suggest that
        riverine SOC is less degraded than its marine counterpart, likely due to the relatively short residence time of SOC in rivers
        (Repasch et al., 2021; Hilton et al., 2015), as compared to SOC in shelf sediments (Bröder et al., 2018). When one looks at

patterns across different pools such as DOC versus SOC or POC versus DOC, the patterns are more ambiguous. This is likely
        caused by (i) a variety of processes such as leaching, sorption or fractionation that are at play between these pools, in addition
        to (ii) the temporal aspect that is different for DOC and POC (daily to seasonal snapshots) than SOC (integrating several years
        to decades). Generally, however, we can say that SOC and POC appear relatively fresh (despite having a high radiocarbon
        age) and DOC appears more degraded (yet with a lower radiocarbon age), as is also found in previous studies (Feng et al.,

2017; Goñi et al., 2000; Salvadó et al., 2016; Tesi et al., 2014; Vonk et al., 2013, 2010). The SOC pool is the main (temporary)
        storage place for permafrost thaw-derived OC, and we propose to devote more scientific attention to the physical and chemical
        processes affecting the transport and degradation of this fraction, as eventually this will determine the fraction of permafrost
        OC that can be captured for long term-burial.

        **4 Conclusions and outlook**

The aim of this study was to use an integrated approach to investigate the changes of different phases of OC (dissolved,
        particulate and sediment OC), and the effect of fractionation and degradation on permafrost-derived OC during transport over
        large distances along the land-ocean continuum. We conclude that permafrost-derived OC makes up the bulk of the total SOC
        along the source-to-sink system, and accounts for a significant part of the POC in the Kolyma River. In contrast, the
        contribution of permafrost-derived OC is marginal to POC in the marine realm and to DOC across the entire transect, despite

the presence of [14]C depleted sources within the watershed. Overall, this highlights the importance of accounting for all carbon
        pools in order to allow for comparisons between fluvial and marine systems across different temporal scales.
        We found a decrease in OC and terrigenous biomarkers normalised to sediment mineral surface area across the transect,
        indicating loss through degradation of terrestrial OC over transport distance. Molecular biomarker proxies indicating OC
        degradation show a remarkably "fresh" biomarker signature for SOC, despite its generally lower $\Delta^{14}C$ values (i.e. older) than

DOC and POC. Biomarker degradation proxies along the land-ocean continuum generally compare well between river samples
        and marine samples, yet show diverse degradation patterns when comparing between different OC pools (e.g., DOC versus
        SOC). Processes such as leaching and sorption, causing transfer of OC between DOC and POC pools, may explain some of
        the patterns we observed, in addition to the contrasting timescales that these pools represent (from days to years). We therefore
        want to emphasize that an integrated approach is necessary to obtain a complete picture of OC transport along the river-ocean

continuum, and recommend to (a) minimally compare one pool (e.g., SOC) across land-ocean transects, and ideally (b)
        compare all pools (SOC, POC, DOC) consistently across land-ocean transects. Furthermore, as we here have shown that
        permafrost-derived OC is mostly transported within the SOC fraction, we recommend to increase scientific focus on the
        sedimentary fraction when studying the fluvial and marine fate of permafrost OC.
        Finally, we want to acknowledge that large discrepancies remain between the freshwater and marine research fields when

studying OC dynamics as marine studies seem to focus mostly on SOC, while river studies mostly target DOC.  Linking these





two environments and applying common methodology is necessary to completely resolve the fate of terrestrial OC along river-shelf systems.

**Appendix**

**Table A1. East Siberian Sea (ESS) sample locations, names, and distance from the mouth of the Kolyma River for sediment samples, dissolved organic carbon samples (DOC) and particulate organic carbon samples (POC). Data used in this study was gathered from four earlier publications: Bröder et al., 2019; Salvadó et al., 2016; Tesi et al., 2014; Vonk et al., 2010. Specifically, Tesi et al., 2014 and Vonk et al. 2010) have characterized surface water DOC and POC in the ESS, along with underlying surface sediments, following the paleo river valley of the Kolyma up to 600 km offshore, collected on 3-5 September 2008, and data from a more recent cruise (between 31 July and 4 August 2014) are used to extend this transect up to 1000 km offshore (Bröder et al., 2019; Salvadó et**
**al., 2016).**

| Sample name/site | Latitude (°) | Longitude (°) | Distance from river mouth (km) | References for data |
|---|---|---|---|---|
| **Sediment** | | | | |
| YS034B | 69.71 | 162.69 | -50 | Tesi et al., 2014; Vonk et al., 2010 |
| YS035 | 69.82 | 164.06 | -100 | Tesi et al., 2014; Vonk et al., 2010 |
| YS036 | 69.82 | 166.00 | -170 | Tesi et al., 2014; Vonk et al., 2010 |
| YS037 | 70.13 | 168.01 | -260 | Tesi et al., 2014; Vonk et al., 2010 |
| YS038 | 70.70 | 169.13 | -330 | Tesi et al., 2014; Vonk et al., 2010 |
| YS039 | 71.22 | 169.37 | -390 | Tesi et al., 2014; Vonk et al., 2010 |
| YS040 | 71.48 | 170.55 | -440 | Tesi et al., 2014; Vonk et al., 2010 |
| YS041 | 71.97 | 171.79 | -510 | Tesi et al., 2014; Vonk et al., 2010 |
| YS086 | 75.30 | 174.40 | -760 | Tesi et al., 2014; Vonk et al., 2010 |
| YS088 | 75.10 | 172.19 | -700 | Tesi et al., 2014; Vonk et al., 2010 |
| YS090 | 74.67 | 172.39 | -660 | Tesi et al., 2014; Vonk et al., 2010 |
| YS091 | 74.43 | 170.85 | -620 | Tesi et al., 2014; Vonk et al., 2010 |
| SWE-60 | 73.52 | 169.46 | -510 | Bröder et al., 2019; Salvadó et al., 2016 |
| SWE-61 | 74.11 | 170.90 | -580 | Bröder et al., 2019; Salvadó et al., 2016 |
| SWE-63 | 74.68 | 172.37 | -660 | Bröder et al., 2019; Salvadó et al., 2016 |
| SWE-64 | 74.94 | 172.69 | -700 | Bröder et al., 2019 |
| SWE-65 | 75.16 | 173.19 | -720 | Bröder et al., 2019 |
| SWE-66 | 75.90 | 174.30 | -810 | Bröder et al., 2019; Salvadó et al., 2016 |
| SWE-67 | 76.32 | 175.61 | -860 | Bröder et al., 2019; Salvadó et al., 2016 |
| **DOC** | | | | |
| SWE-60 | 73.52 | 169.46 | -510 | Salvadó et al., 2016 |
| SWE-61 | 74.11 | 170.90 | -580 | Salvadó et al., 2016 |
| SWE-63 | 74.68 | 172.37 | -660 | Salvadó et al., 2016 |
| SWE-66 | 75.90 | 174.30 | -810 | Salvadó et al., 2016 |



**Table A1. (cont.)**

| Sample name/site | Latitude (°) | Longitude (°) | Distance from river mouth (km) | References for data |
|---|---|---|---|---|
| **POC** | | | | |
| YS-34B | 69.71 | 162.69 | -50 | Tesi et al., 2014; Vonk et al., 2010 |
| YS-35 | 69.82 | 164.06 | -100 | Tesi et al., 2014; Vonk et al., 2010 |
| YS-36 | 69.82 | 166.00 | -170 | Tesi et al., 2014; Vonk et al., 2010 |
| YS-37 | 70.13 | 168.01 | -260 | Tesi et al., 2014; Vonk et al., 2010 |
| YS-38 | 70.70 | 169.13 | -330 | Tesi et al., 2014; Vonk et al., 2010 |
| YS-39 | 71.22 | 169.37 | -390 | Tesi et al., 2014; Vonk et al., 2010 |
| YS-40 | 71.48 | 170.55 | -440 | Tesi et al., 2014; Vonk et al., 2010 |
| YS-41 | 71.97 | 171.79 | -510 | Tesi et al., 2014; Vonk et al., 2010 |
| SWE-60 | 73.52 | 169.46 | -510 | Salvadó et al., 2016 |
| SWE-61 | 74.11 | 170.90 | -580 | Salvadó et al., 2016 |
| SWE-63 | 74.68 | 172.37 | -660 | Salvadó et al., 2016 |
| SWE-66 | 75.90 | 174.30 | -810 | Salvado et al., 2016 |

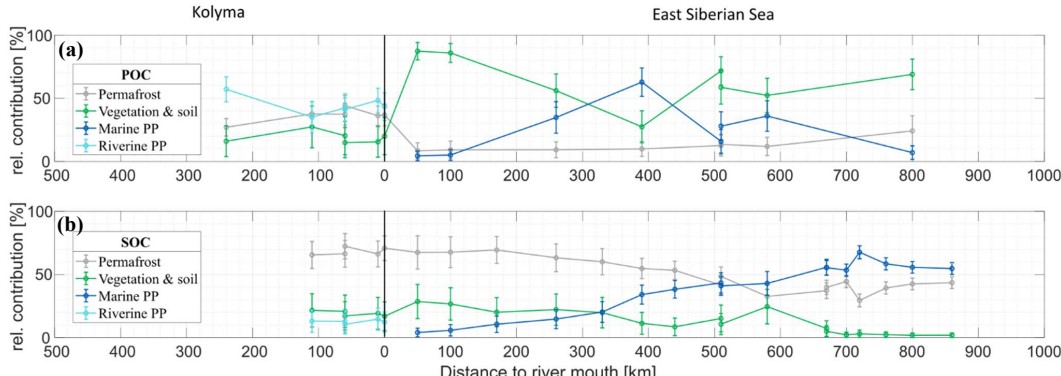

**Figure A1. Relative contribution of three endmembers over transect distance for (a) surface water particulate organic carbon (POC) and (b) sediment organic carbon (SOC), based on dual carbon isotope ($\Delta^{14}$C and $\delta^{13}$C) endmember analyses (EMMA). For the riverine part of the transect (Kolyma; left side of the figure), the endmembers are: Permafrost organic carbon (OC) in grey, Vegetation/soil OC in green, and Riverine primary production (PP) OC in cyan. For the marine part of the transect (East Siberian Sea; right side of the figure), the endmembers are: Permafrost OC in grey, Vegetation/soil OC in green, and Marine primary 590 production OC in dark blue. This figure was made using the alternative Marine PP endmember ($\delta^{13}$C = -21 ± 1‰), for comparison to Figure 4 of the main text, which uses the Marine PP endmember of -24 ± 3‰).**



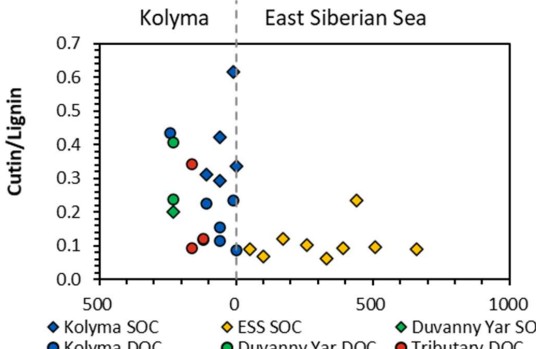

**Figure A2. Cutin to Lignin ratios of Kolyma (blue), Duvanny Yar (green), tributaries (red), and East Siberian Sea (ESS; yellow) sediment organic carbon (SOC; diamonds) and dissolved organic carbon (DOC; circles).**

**Data availability**

All data that support the findings of this study are included within the article and/or are available for download in earlier publications.

**Author contributions**

Conceptualisation: JEV, DJ; Formal analysis: DJ, LB, TT, PP, NH; Funding acquisition: JEV; Investigation: DJ, LB, KK, AD,
NZ; Resources: TT, NZ, AD, NH, TE, JEV; Writing – original draft preparation: DJ; Writing – review & editing: all authors.

**Competing interests**

The authors declare that they have no conflict of interest.

**Acknowledgements**

This project was funded by the European Research Council as a Starting grant to J.E. Vonk (THAWSOME #676982). We
want to thank the owners and staff of the Northeast Science Station (Cherskiy, Russia) for their logistical support during the field campaign. We want to thank Suzanne Verdegaal-Warmerdam, Oscar Kloostra, Martine Hagen and Roel van Elsas of the VU Amsterdam Sediment Lab and Stable Isotope Lab, and the staff of the ETH Laboratory of Ion Beam Physics for their help with the laboratory analyses.

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
