# Peer review of "Contrasts in dissolved, particulate and sedimentary organic carbon from the Kolyma River to the East Siberian Shelf"

_EGUsphere, 2022_

## Author Comment (AC1)

EGUsphere, referee comment RC1
https://doi.org/10.5194/egusphere-2022-516-RC1, 2022

[Figure]

**Comment on egusphere-2022-516**
*With author response added in blue italics*

Anonymous Referee #1
* * *
Referee comment on "Contrasts in dissolved, particulate and sedimentary organic carbon from the Kolyma River to the East Siberian Shelf" by Dirk J. Jong et al., EGUsphere, https://doi.org/10.5194/egusphere-2022-516-RC1, 2022
* * *
Jong et al. offer a look at the organic carbon (OC) pools across the rapidly changing land-ocean interface associated with the Kolyma River. They use stable and radio isotopes of OC as well as lignin, phenol and lipid biomarkers to demonstrate how the composition of the dissolved (DOC), suspended particulate (POC) and sedimentary (SOC) OC pools varied from Kolyma River tributaries, along the river itself and out to the East Siberian Sea. They find considerable variability in the age and sources of OC between these pools and across their transect.

Overall, I enjoyed reading the manuscript. It is clearly written and importantly, integrates terrestrial (freshwater) and marine measurements along a coastal margin experiencing rapid environmental change, something that is still not often done. I do believe that the manuscript will be of interest to the readers of Biogeosciences, particularly those interested in carbon dynamics at the coastal margin and/or in Arctic regions. The most substantive changes that I have suggested relate to the inclusion of more information in the Methods section to enable replicability of the methods, particularly as it relates to sample processing and analysis. Note also one editorial suggestion related to the description of isotope ratios that will involve a detailed look throughout the manuscript.

*Thanks a lot for your review and comments. We are glad to hear you enjoyed reading the manuscript and believe it will be interesting for the readers of Biogeosciences.*

**General considerations for the authors**:

- Until the reader consults Vonk et al. (2012), Wild et al. (2019) and Bröder et al. (2020), how this study differs from those published previously by some of the author team is not entirely clear. It would be helpful for the reader if this was made more explicit in the manuscript itself. There is obviously great value in using previously published data to answer new questions, but what distinguishes this study could be more clear.

*We will make it more clear, earlier on in the manuscript, that new analyses presented in our manuscript are only from the Kolyma River, yet we compare with published results from the shelf. This combined, integrated, approach is the novelty of this study; looking at all three carbon species that are transported, and in combining a riverine transect and a marine transect. For example, Vonk et al. (2012) looks at SOC on a marine transect, starting from the Kolyma Delta, Wild et al. (2019) looks at riverine DOC and POC at one location in the Kolyma river, same as Bröder et al. (2020), who are looking at the temporal resolution at one location in the Kolyma and one tributary. In this paper we instead integrate all these samples, and view the river, the delta and the shelf as a continuous system. This comment ties in with the first comment of the other reviewer, so it is a priority in our revisions.*

▪ There is a temporal offset between the collection of the riverine (2018) and marine (2008 & 2014) data. Do the authors think that this temporal offset could be important? Did anything important happen within the watershed during that time that might be reflected in the organic carbon pool? In 2021, for example (evidently outside of the sampling time period but likely not an isolated incident), widespread wildfires occurred within the Kolyma River watershed. Wildfires are just one example of events that are known to impact both permafrost thaw dynamics but also organic carbon pools.

*We acknowledge there is a temporal offset, which is logistically difficult to avoid in these remote settings. We have now mentioned this more explicitly. We do believe that the impact of this offset is limited as (i) there are – as far as we could find – no extreme events during those years, and (ii) the sampling of our campaign and the two marine campaigns fits hydrographically to how river constituents flow (i.e. river sampling end of July/August, shelf waters beginning of September), all after spring freshet, without presence of sea ice. There certainly is some interannual variability for DOC and POC (see Bröder et al. (2020) and McClelland et al (2016) for example), but when averaged out for the same season over the years no big changes are expected. For SOC, the residence time within the river can be in the order of decades, so this temporal offset plays only a limited role.*

**Additional details in the method**:

▪ Section 2.1: Could more information on the receiving ocean environment be provided? Is the Kolyma River at station K6, for example, tidally influenced? How do waters circulate within the East Siberian Sea? How was from the edge of the continental shelf from the sampling site furthest from land.
*We will add a few lines on this to section 2.1. Tidal influence is minimal (tidal range in order of decimeters), K6 was fresh water, YS36 saline water, and the edge of continental shelf was close to the furthest sampling point (Bröder et al., 2019). We added water depth to Figure 1 and Table A1 to illustrate that all sediment samples were taken on the shelf, with the final two samples (SWE-66 and SWE-67) close to/at the shelf break at 239 and 468 m water depth. We also added some information on the receiving ocean (salinity, dominant oceanic currents) to the methods section.*

▪ Were replicates collected/run for any of the analyses?
*We did not run replicates in this study, due to scarcity of the sampled material. However, standards were run alongside all samples and outliers were re-run to ensure representative analyses.*

▪ L178-179: How many subsamples were collected for each sample?
*From the filters one subsample was punched out for POC/POC13C and one for POC14C. For clarification, we changed this line to: "A subsample was punched out of each 47 mm GF/F filter, placed in a pre-combusted silver capsule, and weighed." And line 189 to "A*

second subsample of the GF/F filters (POC) was punched-out and a subsample of sediment (SOC) was taken and weighed in pre-combusted silver capsules for $^{14}C$ analyses."

- L189: How were the filters subsampled for the radiocarbon analyses?
  *We used the same method as for POC13C, see comment above.*

- L211: How was it determined to select one, two or three GF/F filters for the analysis?
  *The expected amount of OC on the filters was calculated using the POC concentration and the volume of water filtered. Some filters did not contain enough material for the analyses, so multiple filters of that location were extracted. We added: "For some locations, multiple filters (up to three) had to be extracted to contain enough material (~6 to 26 mg OC)."*

- I'd be curious to know why two different acidification techniques (direct acidification and fumigation) were used to remove inorganic carbon from the SOC/POC samples for the stable isotope and radioisotope analyses, respectively? Do the authors have confidence that the two were equally effective in removing inorganic carbon?
  *As samples from this region are moderately low in CaCO3 (<3% $CaCO_3$), we are confident that these methods are equally efficient in removing inorganic C. For example Komada et al. (2008) show IC removal rates of 99.4 ±0.2% for $HCl_{aq}$ and 99.5 ±0.1% for $HCl_{vap}$.*

- Were the samples for stable isotope analysis rinsed or neutralized following HCl addition?
  *They were neutralized by drying at 60°C in a desiccator with a dish of NaOH pellets on the bottom. We have added this information to the method section (line 185).*

- L238-240 and L247-251: How many samples contributed to the mean +/- SD used for the permafrost OC and primary production end-member values?
  *We added the n = x to the permafrost endmember based on Wild et al. (2019).*

- L245-247: Why did the authors choose to include vegetation and soil OC as one end-member instead of two as in the source publication?
  *The source publication was in a marine setting, that did not consider autochtonous riverine production, while we instead want to include this. Since we only have two endmember parameters in our model ($\delta^{13}C$ and $\Delta^{14}C$), a maximum of three OC sources can be apportioned. We chose to combine the other potential sources of OC that lie in a similar isotopic range: modern vegetation and soil OC as well as those of Pleistocene and Holocene permafrost.*

- Please specify whether means and standard deviations are indicated throughout the manuscript or if not, what metrics of average and variance are used.
  *Means and standard deviation are used throughout the manuscript. This is now correctly indicated in the relevant places.*

- It is not clear until Table 2 that SOC was not collected at all sites.
  *We now clarify that in the methods under 2.2.2: "Riverbed sediments of the Kolyma main stem were sampled using a Van Veen grab-sampler, …"*

- L260 – 263: How was convergence of the Bayesian model assessed?
  *Several samples/sites in our study where already analyzed in the past (see Vonk et al. 2010), which helped tuning the number of necessary iterations to achieve convergence, reflecting the known apportionment. The other samples/sites were then modelled using the same number of iterations, but each sample/site was ran up to 10 times checking that the variability did not become too large. The chosen model parameters (1,000,000 iterations, a burn-in (initial search phase) of 10,000, and a data thinning of 10) converged sufficiently to a stable apportionment.*

**Figures and figure captions**:

- Figure 1 caption: Include the names of the tributaries and their abbreviations as a key in the caption.
  *Good suggestion, the names of the tributaries were added to the caption.*
  *.*

- Figure 1c: It may be helpful for the reader to change the colours of marine sampling stations to distinguish between the two sampling campaigns. If possible, it would be great to see a bathymetry layer added to the figure, which would help in describing the receiving marine environment (as above).
  *We've added the bathymetry to the figure, and adjusted the colours of the marine sampling station as suggested by the reviewer. The sampling stations from the 2008 cruise remained red, the points from the 2014 cruise were made green in the figure.*

- Figure 2 caption: Indicate whether the "average" refers to a mean or median.
  *We indicated that the values are mean ± standard deviation.*

- Figure 4 caption: What values are presented in the figure? Are these Bayesian median credible intervals? Means?
  *We agree with the reviewer that this should have been written in the figure caption and this will be accordingly corrected in the revised manuscript. The presented values are indeed mean ± standard deviation from Monte Carlo simulations. They have been added to show the uncertainty associated with Bayesian modelling.*

- Figure 5 caption: It does not appear as though any POC samples have been included in the figure, though POC (triangles) is included in the caption.
  *Thanks for noticing this. We removed POC from the caption.*

- Possible additional supplementary figure: It might be helpful to include a hydrograph of the Kolyma River as a supplementary figure and outline the time period over which the presented samples were collected. This would help to give hydrological context to the samples presented.
  *We added a figure of the discharge of the Kolyma River as Figure A1, and added a reference to the figure in section 2.1. The sampling of this study took place directly after spring freshet peak discharge.*

**Results and Discussion:**

- L277: Indicate the range of DOC concentrations observed in ESS surface waters from the literature for the reader to be able to make the comparison.
  *Thank you for pointing to this, we have now changed it to: "The DOC concentrations along the Kolyma River transect range from 2.76 to 4.97 mg L$^{-1}$, which is a bit higher than DOC in ESS surface waters (~0.6 – 1.8 mg L$^{-1}$; Salvado et al., 2016; Alling et al., 2010)."*

**Editorial changes**:

- L134: Change "or" to "of"
  *Removed "or the fastest flowing part of the river.", as only one sample was not in the center, which is already specified in the text.*

- L182: Remove "in" between "acidified" and "as described".
  *Done*

- L246: Change "weighed" to "weighted".
  *Done*

- Throughout the manuscript (example usages on L324, 334, 335, 336, 342, etc.): This is a matter of semantics, but $\delta^{13}C$ and $\Delta^{14}C$ are ratios and the ratio itself cannot inherently be more enriched or depleted. For $\delta^{13}C$, for example, the sample is more enriched or depleted in the heavier isotope $^{13}C$ or in the lighter isotope $^{12}$ Alternatively, the ratios can be described as higher or lower, but not enriched or depleted without specifying to which of the two isotopes these modifiers refers. See the guide to Common Mistakes in Stable Isotope Terminology and Phraseology: http://dx.doi.org/10.6084/m9.figshare.1150337
  *We agree with the reviewer that our δ13C and Δ14C terminology was not always correct, and we have corrected this throughout the manuscript.*

---

## Author Comment (AC2)

EGUsphere, referee comment RC2
https://doi.org/10.5194/egusphere-2022-516-RC2, 2022

[Figure]

**Comment on egusphere-2022-516**
**With author response added in blue italics**

Anonymous Referee #2
* * *
Referee comment on "Contrasts in dissolved, particulate and sedimentary organic carbon from the Kolyma River to the East Siberian Shelf" by Dirk J. Jong et al., EGUsphere, https://doi.org/10.5194/egusphere-2022-516-RC2, 2022
* * *
The Jong et al. manuscript contained an enriched dataset of organic matter in different forms, including dissolved, suspended and sedimentary, from samples collected along the Kolyma River to the East Siberian Shelf. A comprehensive list of parameters was measured on these samples, including carbon stable and radio-isotopes, lignin phenols, lipid biomarkers, mineral specific surface area etc. They also used a mixing model to quantify the contribution of organic matter from three endmembers to these samples. The main conclusion was that DOC, POC and SOC along the transect have distinct compositional and degradation patterns, with significant contributions from permafrost- derived OC, particularly for SOC and DOC. It was also concluded that degradation occurred along the river to ocean transit based on biomarkers and OC loadings on minerals, among other minor conclusions.

Clearly this data is much more comprehensive than what has been published about the Kolyma River, or other Arctic rivers in general, as they included all three phases of organic carbon, and bulk and specific parameters. These data will be of value to the community, thus need to be published. The conclusions are solid, although I have to say that they are kind of expected and it is hard to find anything particular novel from what we already know.

*The novelty of our study lies in the combination of the extensive fluvial dataset (three OC components along a river transect) with existing marine shelf/water column data. This shows that the largest shifts in OM composition actually occur in between the fluvial and the marine realm. We have highlighted this finding and its implications better in the manuscript.*

It is great that DOC, POC and SOC were all measured in a same study, but the authors need to acknowledge the fact that SOC may be in totally different time scales in terms of mobilization and transport than DOC and POC. DOC and POC are co-transported with water flow, but SOC is likely not unless in a storm fasion. In other words, their resience times are way different.

*We agree that we should put more emphasis on this, as now it is only mentioned in the final paragraph of the discussion. We will add information on this matter in appropriate places in the revised manuscript.*

It is also not clear the depth of riverbed sediment was collected. This is important to know, as one could imagine surface 1cm could be very different from 10cm, in terms of not only the transport but also the level of dissolved oxygen which would affect degradation. The authors need to factor this in to the text.

*We sampled the first couple of cm (approximately 1 – 5 cm) of surface sediment from the riverbed. The Van Veen grab sampler is not the most consistent device, especially using it in fast flowing river water, but any sample we obtained that looked 'intact' and of sufficient volume (i.e. no sample running out/leaky sampler) was stored and analyzed.*

*We changed the first sentence of 2.2.2 to: "Riverbed sediments were sampled using a Van Veen grab-sampler, sampling surface sediment up to 1 – 5 cm, and stored in sterile Whirl-Pak® bags."*

Despite the comprehensives of this dataset, I still feel that there are a couple of key parameters missing, which would strengthen their arguments. For example, production was attributed to be the major contributor to the POC, but why not directly quantify the Chla concentration? This would direct address riverine production. 14C-DOC was not measured, either. They offered a couple of references, but I think this is a key parameter to have, particularly because its changes along the transect would offer further insights into the OC dynamics. The situation may not be as simple as cited, "earlier studies show that Kolyma River and tributary DOC is relatively young…". Similarly, I am not sure why lignin phenols were not measured on POC?? This would directly address the contribution of terrestrial plants…

*Thanks, we agree that these parameters would have been valuable to measure. As our initial focus was not so much riverine production (but instead tracing terrestrial matter) we did not collect Chlorophyll a. In hindsight this would have been very valuable.*

*For DOC-14C and lignin POC there were some methodological constrains;*

*For DOC-14C, our solid phase extraction setup (to concentrate DOC) was not guaranteed 14C-contamination proof, and we had limited space to bring whole water samples for DOC 14C back. Besides, there really are quite a few studies that have measured 14C-DOC in the Kolyma mainstem, and our choice was to focus more on the composition.*

*For POC lignin, unfortunately the GFF filters we used to collect the material could not be used in the lignin extraction protocol. A different type of filter would be needed to collect samples for quantification of lignin phenols.*

One of the motivations for conducting this work was the elusive nature of cycling and degradation of POD during the lateral transport through the whole watershed, as set up in the Introduction by the authors. However, when all the data are integrated, say from Figures 3-7, the degradation signals were most pronounced from the river mouth to East Siberian Sea, regardless of the end member contribution (Fig. 4), normalized biomarker centration (Fig. 6), or biomarker degradation (Fig. 7). In a sense, I think that these data collectively mean that the estuary section is more important than the river stream itself in terms of organic matter processing. Yet, this was not discussed but should be (even though you may not agree with me).

*We fully agree with the reviewer on this point, and see that this comment ties in with the first comment (the novelty of this study). By integrating data on all carbon species and by presenting the data as a continuous transect, we see that the transition zone between river and ocean is the place where most changes happen. This will be included in the next revision of the paper, as in the response to the first comment.*

Line 60: delete the "." before "degradation"

*Thanks for noticing this!*

Line 68: should be "Hilton et al. (21015)"

*Changed to "Hilton et al. (2015)".*

Line 121-130: it is a bit awkward to have a table and figure in the introduction. I would suggest that this be moved to the next section.

*This figure and table are under section "2.1 Study area and sample locations" in the chapter "Methods", which is the appropriate location for it according to us.*

Line 153: how deep did the sampler penetrate? This may be important information (see my comment above).

*The Van Veen sampler sampled the top 1-5 cm of surface sediment, we have added this to the text (see full response above).*

Line 174: change to "according to Deirmendjian et al. (2020)."

*Changed to "following the method of …"*

Line 252: it's not clear what you meant by "…our own algal sample". How do you know it was algal bloom? And there would be other types of organic matter in a riverine sample!

*This was a visual observation while sampling. We cannot rule out a small contribution of terrestrial OC here, but given the very high OC concentration of the particulate matter sampled here (46%) points towards it consisting of almost pure, likely algal, organic matter.*

*We changed this sentence to "… and the sample of  the Panteleikha River from this study ($\delta^{13}C$ = -33.5‰, $\Delta^{14}C$ = -26‰), where an algal bloom was observed during the study period." To clarify a bit better where this statement comes from.*

Line 499: it could be simply due to the conversion of aldehyde to acid during oxidation, not necessarily selective degradation.

*Thank you for this comment. We looked into this again, and think that it may lie in between a "conversion of aldehydes to acids" and "aldehydes degrade faster than acids" as we state the manuscript.*

*To quote Opsahl & Benner (1995): "Elevated Ad/Al ratios are indicative of microbial oxidation of propyl side chains of lignin which increases the carboxyl content of the remaining lignin … "  This means that due to this microbial oxidation process more degraded lignin yields less aldehydes relative to acids, increasing the acid/aldehyde ratio, as lignin is a complex organic polymer and the CuO oxidation process splits the lignin polymer into individual phenols.*

*Taking these things into consideration we will now rephrased this line into: "More degraded lignin yields more acids relative to aldehydes in the CuO extraction process, which is reflected in a higher Vd/Vl and Sd/Sl ratio."*

---

## Author Response (AR2)

*Author response* **to the comments to the author**:

Dear Jong et al.,

Thank you for submitting the revised manuscript. The manuscript has been re-evaluated by a previous reviewer who was overall satisfied with the revisions. I agree that the revisions have addressed the reviewer's concerns and improved the manuscript. The reviewer asked for a couple of minor edits which I think should be easily incorporated in the final version.

I am therefore happy to inform that your manuscript is accepted for publication in Biogeosciences pending the final corrections requested by the reviewer (see below).

*Thank you! We are thrilled to have our manuscript accepted for publication in Biogeosciences.*

Reviewer comments:
Thanks so much for addressing all the comments. I requested information about algal production in the river, possibly some Chla measurement, but they did not have such data. I am wondering whether they could add some references about Chla data in rivers in this region or others in the Arctic, so readers have a general idea about the productivity in this river.

*We have added a reference to a phytoplankton study in the Lena Delta to line number 353: "This abrupt transition between fresh water and saline water POC composition is likely tied to the different phytoplankton communities present in these respective environments, as seen in other Arctic river deltas (e.g. Kraberg et al., 2013; Lena River delta)."*

| | |
|---|---|
| Line 82, should be: "...2016)." | *Edited* |
| Line 85 delete the "." after degradation. | *Edited* |

Thank you for submitting your work to Biogeosciences.

Best regards,
Yuan Shen
Associate Editor